# A novel mechanism for dissimilatory nitrate reduction to ammonium in *Acididesulfobacillus acetoxydans*

Reinier A. Egas,[1] Julia M. Kurth,[1,2] Sjef Boeren,[3] Diana Z. Sousa,[1,4] Cornelia U. Welte,[5] Irene Sánchez-Andrea[1,6]

**ABSTRACT** The biological route of nitrate reduction has important implications for the bioavailability of nitrogen within ecosystems. Nitrate reduction via nitrite, either to ammonium (ammonification) or to nitrous oxide or dinitrogen (denitrification), determines whether nitrogen is retained within the system or lost as a gas. The acidophilic sulfate-reducing bacterium (aSRB) *Acididesulfobacillus acetoxydans* can perform dissimilatory nitrate reduction to ammonium (DNRA). While encoding a Nar-type nitrate reductase, *A. acetoxydans* lacks recognized nitrite reductase genes. In this study, *A. acetoxydans* was cultivated under conditions conducive to DNRA. During cultivations, we monitored the production of potential nitrogen intermediates (nitrate, nitrite, nitric oxide, hydroxylamine, and ammonium). Resting cell experiments were performed with nitrate, nitrite, and hydroxylamine to confirm their reduction to ammonium, and formed intermediates were tracked. To identify the enzymes involved in DNRA, comparative transcriptomics and proteomics were performed with *A. acetoxydans* growing under nitrate- and sulfate-reducing conditions. Nitrite is likely reduced to ammonia by the previously undescribed nitrite reductase activity of the NADH-linked sulfite reductase AsrABC, or by a putatively ferredoxin-dependent homolog of the nitrite reductase NirA (DEACI_1836), or both. We identified enzymes and intermediates not previously associated with DNRA and nitrosative stress in aSRB. This increases our knowledge about the metabolism of this type of bacteria and helps the interpretation of (meta)genome data from various ecosystems on their DNRA potential and the nitrogen cycle.

**IMPORTANCE** Nitrogen is crucial to any ecosystem, and its bioavailability depends on microbial nitrogen-transforming reactions. Over the recent years, various new nitrogen-transforming reactions and pathways have been identified, expanding our view on the nitrogen cycle and metabolic versatility. In this study, we elucidate a novel mechanism employed by *Acididesulfobacillus acetoxydans*, an acidophilic sulfate-reducing bacterium, to reduce nitrate to ammonium. This finding underscores the diverse physiological nature of dissimilatory reduction to ammonium (DNRA). *A. acetoxydans* was isolated from acid mine drainage, an extremely acidic environment where nitrogen metabolism is poorly studied. Our findings will contribute to understanding DNRA potential and variations in extremely acidic environments.

**KEYWORDS** DNRA, nitrite reduction, acidophilic sulfate-reducing bacteria, acid mine drainage, *asrABC*, nitrosative stress, NirA, transcriptome, proteome

O ur insight into the biogeochemical nitrogen cycle has increased in recent years with the discovery of numerous new microbial nitrogen-transforming redox reactions and pathways (1). Under oxygen limitation, two reactions in the nitrogen cycle control the bioavailability of nitrogen in its most oxidized form, nitrate ($NO_3^-$) (2). These are denitrification, the reduction of nitrate to dinitrogen gas ($N_2$), and the

Address correspondence to Irene Sánchez-Andrea, irene.sanchez@ie.edu.

The authors declare no conflict of interest.

See the funding table on p. 15.

dissimilatory nitrate reduction to ammonium, (DNRA). DNRA is ecologically relevant to avoid nitrogen loss in nitrogen-depleted environments. In the last decade, DNRA has proven to be remarkably versatile, with novel mechanisms identified and even additional intermediates such as hydroxylamine ($NH_2OH$) proposed (3–8). Studying microorganisms performing DNRA while lacking classical DNRA enzymes could reveal alternative reactions and enzymes involved in DNRA. This will increase the understanding of both nitrogen metabolism and the nitrogen cycle (1, 2).

DNRA canonically proceeds via two steps: the initial two-electron reduction of nitrate to nitrite, followed by the six-electron reduction of nitrite to ammonia. Novel mechanisms were recently identified for both steps. For the first step, two canonical membrane-bound dissimilatory nitrate reductases are recognized, the Nap- and Nar-type nitrate reductases (9). The periplasmic-oriented Nap-type (NapAB) has its active site in the periplasmic NapA subunit, whereas the bacterial cytoplasmic-oriented Nar-type (NarGHI) has its active site in the cytoplasmic NarG subunit (10, 11). Recently, a third dissimilatory nitrate reductase has been identified, the cytoplasmic nitrite oxidoreductase multiprotein complex (NxrABC) (5). However, the precise mechanism through which NxrABC controls the direction of the electron flow is still unclear.

The best described dissimilatory nitrite reductase is the periplasmic NrfAH (12). Recently, three additional dissimilatory nitrite reductases have been identified: (i) the epsilonproteobacterial hydroxylamine oxidoreductase (Hao); (ii) the octaheme nitrite reductase (ONR) from *Thioalkalivibrio nitratireducens*; and (iii) the octaheme tetrathionate reductase (OTR) from *Shewanella oneidensis* (3, 13, 14). For both ONR and OTR, it is unclear if their physiological contribution is to detoxify or to respire either nitrite or hydroxylamine. In addition to these dissimilatory mechanisms, NasB, NirA, and NirB can assimilate nitrite into ammonium during assimilatory nitrite reduction (15, 16). Regardless of their involvement in assimilatory or dissimilatory nitrite reduction, all described enzymes only use nitrite as a free intermediate in DNRA. Hydroxylamine was suggested as an intermediate in DNRA, and although detected during nitrate reduction, it has never been measured as a free intermediate (6). The role of hydroxylamine in DNRA and, therefore, in other redox transformations in the nitrogen cycle remains enigmatic (6, 7). To understand the physiological role and function of candidate genes and enzymes, an in-depth analysis is needed to confirm the utilized mechanisms and produced intermediates during DNRA.

In this study, we elucidate an alternative DNRA pathway present in the acidophilic sulfate-reducing bacterium (aSRB) *Acididesulfobacillus acetoxydans*, isolated from acid mine drainage sediment (17). These are extreme environments given their low pH (<3) and high metal concentrations. *A. acetoxydans* reduces nitrate to ammonia, and while encoding a Nar-type nitrate reductase, it does not encode any of the seven known nitrite reductases (NrfAH, NirA, NirB, NasB, HaoA, ONR, and OTR). To identify the enzymes involved in DNRA, we performed comparative transcriptomics and proteomics with sulfate and nitrate as electron acceptors. We also measured all potential nitrogen intermediates during the cultivation of *A. acetoxydans* while reducing nitrate to ammonia. Resting cell experiments were performed to investigate the biotic transformation of nitrate, nitrite, and hydroxylamine by *A. acetoxydans*. We demonstrate that the DNRA pathway of *A. acetoxydans* proceeds via enzymes not previously associated with DNRA and provide data-driven hypotheses which reactions and enzymes are involved in both DNRA and coping with nitrosative stress.

## RESULTS

### *A. acetoxydans* performs DNRA while accumulating nitrite and nitric oxide

To verify the reduction in nitrate, *A. acetoxydans* was grown in cultures with 10 mM nitrate and 5 mM glycerol (Fig. 1A). Nitrate reduction to ammonia was observed, accompanied by the accumulation of nitrite, which was subsequently almost entirely reduced to ammonia (Fig. 1B). Nitric oxide was measured in nanomolar concentrations throughout the cultivation, increasing to micromolar concentrations at the end of

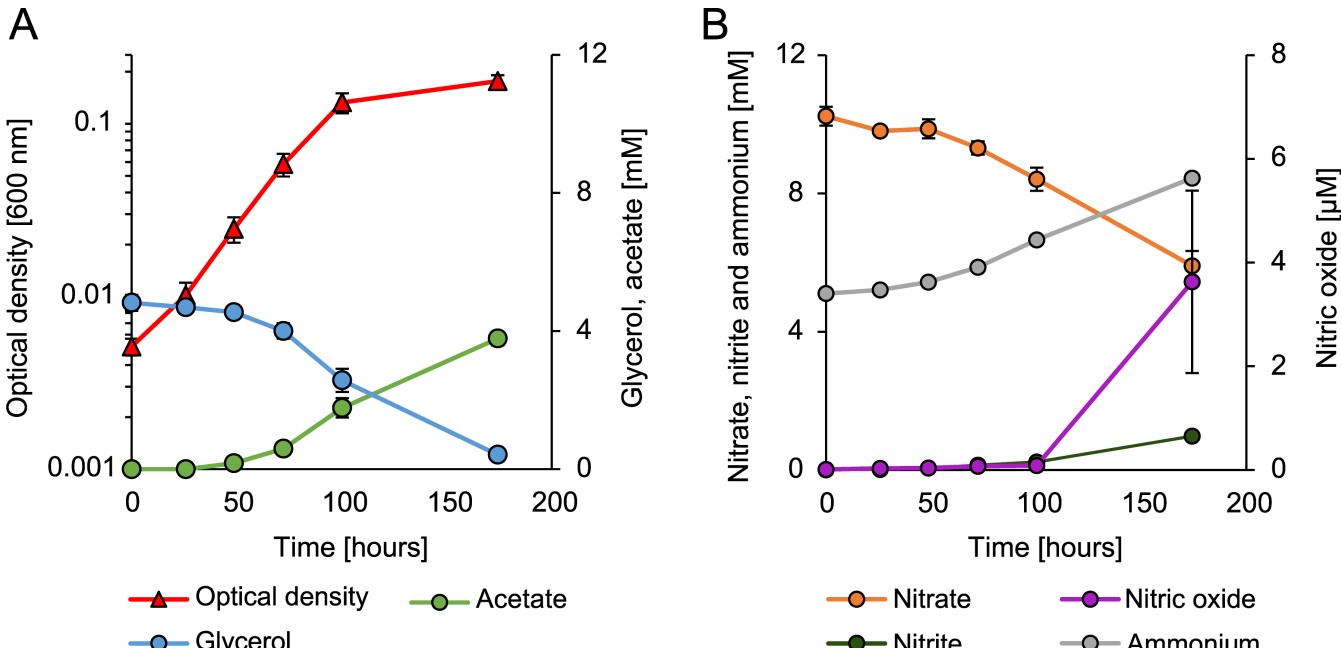

**FIG 1** Growth of *A. acetoxydans* under nitrate-reducing conditions. (A) Growth curve of *A. acetoxydans* correlated to glycerol uptake and acetate production. (B) Nitrogen compound consumption and production during the growth of *A. acetoxydans*. Nitrate is partially reduced to ammonium, while nitrite accumulates; during the cultivation, nitric oxide was produced up to micromolar concentrations. No nitrogen loss was observed, excluding other major end products or intermediates. Error bars represent the SD of biological replicates ($n = 5$).

the cultivation (maximum 3.6 µM). In all nitrate-reducing cultures, both growth and metabolic activity halted at ~0.8–1 mM nitrite, despite abundant glycerol and nitrate present (File S1). At this point, the present resorufin, the reduced form of resazurin, changed bottles' appearance from transparent to pink. This was likely the consequence of nitrite accumulation as it is a potent oxidant. After this color shift, the transfer of the cultures to the fresh medium was unsuccessful (no growth). For the cultivations shown in Fig. 1, a shift in medium color was observed shortly after the last liquid sampling point at 174 hours. From these cultures, another gas phase sample was taken, 76 hours after the last liquid sampling point, which showed a 10-fold increase in nitric oxide from 3.6 to 37.7 µM. No nitrogen loss was observed at any point in the cultivation, excluding additional end products. In these cultivations, yeast extract (0.1 g/L) was present, which contains nitrogenous compounds yet no interference was detected on the nitrogen balances. Chemical controls with 10 mM nitrate showed no nitrite or nitric oxide evolution. However, chemical controls with 10 mM nitrate and 0.5 mM nitrite produced 14 and 25 µM of nitric oxide after 24 and 48 hours, respectively. Concomitantly, nitrite decreased from 468 to 386 and 356 µM, and nitrate increased from 10.16 to 10.25 and 10.35 mM. No increase in ammonia was observed. Hydroxylamine was not detected as an intermediate in any of the cultivations.

## Nitrite and hydroxylamine are reduced to ammonia

Resting cell experiments were performed to study intermediates of nitrate reduction. The resting cell experimental set-up was verified with 1 mM of nitrate, which was nearly completely reduced to ammonia in 9 hours (File S2). After 6 hours, nitrate reduction halted, and the accumulated nitrite was completely reduced to ammonia. No hydroxylamine was observed as intermediate in all tested conditions; due to technical limitations, nitric oxide was not measured. The nitrogen balance was completely closed at each timepoint. The addition of high concentrations of nitrite (1 mM) to the resting cell experiments resulted in no metabolic activity, which is in line with previous

observations for these cultures. The addition of 0.5 mM nitrite resulted in the accumulation of ammonia, confirming that *A. acetoxydans* can reduce nitrite to ammonia (File S2). Although no hydroxylamine was detected as an intermediate, its reduction was tested, and 0.5 mM hydroxylamine was completely reduced to ammonia in 3 hours (File S2). Chemical controls with 1 mM hydroxylamine showed no decrease in hydroxylamine, excluding abiotic reduction. Based on these resting cell experiments, nitrate, nitrite, and hydroxylamine are all reduced by *A. acetoxydans* to ammonia.

## Comparative transcriptomics and proteomics revealed multiple candidates for DNRA

Comparative transcriptomics and proteomics were performed to identify candidate genes and proteins, respectively, for DNRA in *A. acetoxydans*. Cultures of *A. acetoxydans* performing sulfate reduction were used as control. Differentially expressed genes were identified as genes with an expression change of at least fourfold and an adjusted *P* value of ≤0.01 (File S3). Unsurprisingly, no changes were observed in the central carbon metabolism. Under nitrate-reducing conditions, the dissimilatory sulfate reduction pathway was still expressed even though no sulfate was present for over 10 successive transfers. This indicates that the sulfate reduction pathway genes are constitutively expressed in *A. acetoxydans* (Fig. 2). Comparative transcriptomics showed differential expression of 327 genes, of which 163 were upregulated during nitrate reduction.

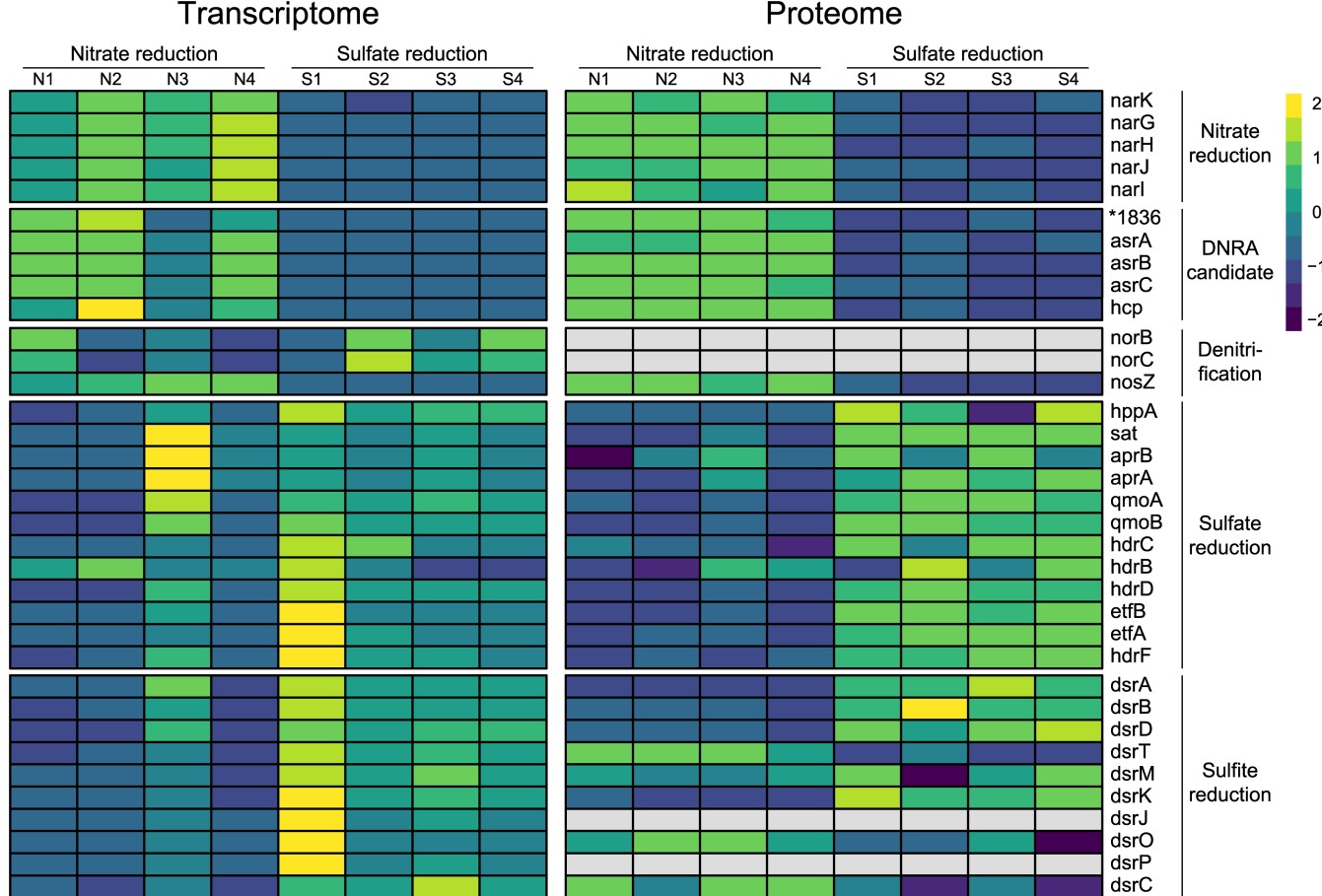

**FIG 2** Heatmaps showing the *A. acetoxydans* comparative transcriptomics with normalized counts (left) and proteomics with label free quantitation (LFQ) values (right) comparing nitrate (N1 to N4) to sulfate (S1 to S4) reduction conditions. Heatmap color codes represent *z*-score normalized values per row comparing across reduction conditions. Depicted genes are grouped by their putative corresponding pathways. Identified candidates are depicted as DNRA candidates. The asterisk indicates the locus tag DEACI. The gray blocks in the proteome (NorBC, DsrJ, and DsrP) indicate that these were not detected.

Nitrate is likely imported into the cell through a putative nitrate/nitrite transporter encoded by the sixfold upregulated *narK* (DEACI_1354). For the first step of DNRA, nitrate reduction to nitrite, NarGHI is most likely responsible as the *narGHI* gene cluster (DEACI_1355-1357) was eightfold upregulated to an average of 204 transcripts per million (TPM).

From the upregulated genes, three cytoplasmic candidates for involvement in DNRA were identified: (i) a putative sulfite reductase (DEACI_1836), whose corresponding gene was upregulated 16-fold with 3,475 TPM; (ii) an anaerobic sulfite reductase *asrABC* (DEACI_4025-4027), whose corresponding genes were upregulated on average 29-fold with 822 TPM; and (iii) hydroxylamine reductase encoded by *hcp* (DEACI_0275), which was upregulated 30-fold with 4,675 TPM. All candidates lack transmembrane helices or twin-arginine translocation system signals, the latter is a part of a signal peptide for protein export.

Comparative proteomics was performed to confirm the translation of expressed candidate genes (File S3). In total, 1,938 proteins were identified, and obtained trends for DNRA candidates aligned with the transcriptome (Fig. 2). Changes in protein abundance of the candidate proteins were even more pronounced than changes in their gene expression. Here, fold change was expressed as the ratio of averaged LFQ values of a protein across all replications between the two conditions. The abundance of the nitrate reductase NarGHI increased 22-fold. Of the identified nitrite transformation candidates, DEACI_1836 increased 29-fold in abundance, AsrABC increased 195-fold, and Hcp increased 52-fold and, strikingly, had the fourth highest LFQ intensity.

Although DEACI_1836 is annotated as a sulfite reductase, the assigned annotation is doubted as a protein homology search could not provide definitive results on its function. DEACI_1836 encodes a 300 amino acid containing protein with two known domains: a nitrite- and sulfite-reductase ferredoxin-like half domain (PF03460) and a nitrite- and sulfite-reductase [4Fe4S] domain (PF01077). The two genes flanking DEACI_1836 encode a hypothetical protein (DEACI_1835) and a ferredoxin putatively containing one [4Fe4S] cluster (DEACI_1837), which were upregulated to 3,500 and 850 TPM, respectively. However, neither was detected in the proteome.

In addition to the cytoplasmic candidates, multiple membrane-embedded proteins potentially involved in the conversion of nitrous oxides were identified, as these were upregulated during nitrate reduction: the NosZDFY and PetABC complexes. These are combined on one operon (DEACI_0711-0719) containing the nitrous oxide reductase NosZ (DEACI_0711), a putative ABC transporter consisting of NosDFY (DEACI_0713-0715), and a three-subunit cytochrome $bc_1$-$c_y$ complex PetABC (DEACI_0717-0719). In this operon, *nosZ* expression increased 127-fold to 1,361 TPM aligning with the proteome, where NosZ increased 423-fold. Although all the protein complexes contain one or more transmembrane helices, except NosY (DEACI_0715) and the cytochrome *b* subunit PetB (DEACI_0718), all were detected at significantly increased abundance in the proteome. The nitric oxide reductase encoded by *norBC* (DEACI_2320-2321) was slightly expressed (<50 TPM) in the transcriptome in all conditions. However, *norBC* did not change in expression level, and neither was NorBC detected in the proteome.

## Upregulation of molybdopterin-containing oxidoreductases

Two gene clusters, DEACI_2560-2564 and DEACI_3268-3271, were upregulated ~13-fold when nitrate was used as the final electron acceptor (File S4). The annotation of either set of genes was not clear, but both operons encode NrfD-like subunit-containing molybdopterin oxidoreductase complexes, which are proteins involved in ion translocation. Functional annotation of these oxidoreductases is difficult, considering the metabolic versatility of the oxidoreductases and the modularity of their subunits (18, 19). They are not identified as nitrite reductase candidates, given their lack of distinct homology to any nitrite reductase, nor are their expression and abundance levels equal to the identified candidates.

## Export and uptake of potential intermediates and end products

Transporters for identified and potential intermediates were detected: uptake of nitrate and export of formed nitrite were likely performed by the nitrate–nitrite transporter NarK (DEACI_1355). Interestingly, a sulfate permease protein (DEACI_0281) was 17-fold upregulated under nitrate-reducing conditions to 605 TPM (although it was not identified in the proteome). For the three potential intermediates, nitric oxide, nitrous oxide, and hydroxylamine, no dedicated transporters were encoded. Formed ammonia, and potentially hydroxylamine, might be exported through the ammonia transporter Amt (DEACI_0023). Upstream there were two highly upregulated hypothetical protein-encoding genes, DEACI_0021 and DEACI_0022, which increased 13- and 16-fold, respectively. Only DEACI_0023 was detected in the proteome, where it significantly decreased under nitrate-reducing conditions.

## Distribution of DNRA candidates in aSRB and neutrophilic type strains

The distribution of identified candidates was inspected in other phylogenetically closely related SRB, with an emphasis on aSRB, through a protein homology search using the protein basic local alignment search tool (BlastP). An identity score ≥30% and a total score ≥100 were implemented as cutoff criteria (Table 1). As a reference, NarG, NapA, NrfA, NirB, AsrA, and Hcp of *Escherichia coli* strain K12 and NosZ of *Paracoccus denitrificans* were used.

Considering aSRB, nitrate and subsequent nitrite reduction was only previously reported for *Thermodesulfobium narugense* (20), *Desulfosporosinus acididurans* (23), and *Ds. acidiphilus* (23), whereas nitrate reduction was reported for *Ds. metallidurans* (24) and *Ds.* sp. I2 (25). Two neutrophilic type strains, *Db. dehalogenans* and *Db. hafniense*, encoded NapA and NrfA in addition to NarG yet only *Db. hafniense* is able to reduce nitrate to ammonia, while *Db. dehalogenans* can reduce nitrate to nitrite without ammonification (36). Interestingly, *Db. chlororespirans* encoded NarG and NapA but is unable to reduce nitrate (39).

DEACI_1836 was identified in all strains, with a >70% identity score in most of the strains. AsrABC was detected in *Ds. acidiphilus*, *Ds. acididurans*, *Ds.* sp. BG, and *Ds.* sp. OT. Only *T. narugense* in this genome comparison is reported as being able to perform nitrite reduction without encoding AsrABC but does encode a putative *nirB*. Unfortunately, for both *Ds.* sp. BG and *Ds.* sp. OT, no physiological data on nitrate or nitrite reduction are available. Interestingly, in none of the neutrophilic strains, AsrA was detected. In contrast to the high Hcp prevalence, NosZ was only detected in *Ds. meridiei*, *Ds. youngiae*, *Ds dehalogenans*, *Ds. hafniense*, and *Db. chlororespirans*. None of these strains were tested for nitric oxide, nitrous oxide, or hydroxylamine production or reduction.

## DISCUSSION

*A. acetoxydans* reduces nitrate forming ammonia as a major end product, with the transient accumulation of nitrite and nitric oxide. While no nitrogen conversion activity was measured in abiotic controls with 10 mM nitrate, supporting the biological character of nitrite formation observed in *A. acetoxydans* cultures, abiotic controls with 10 mM nitrate and 0.5 mM nitrite resulted in concomitant nitrite concentration decrease and increase in nitric oxide and nitrate. Resting cell experiments showed the reduction of nitrate, nitrite, and hydroxylamine to ammonia.

The main question addressed in this study is the conversion of formed nitrite in the absence of known nitrite-reducing enzymes encoded in the genome of *A. acetoxydans*. Nitrite is a potent oxidant and known to be highly reactive (40). Cultivations of *A. acetoxydans* that accumulated 0.8–1 mM nitrite stopped metabolic activity, although nutrients were present in excess. We observed that 0.5 mM nitrite was effectively reduced to ammonia. At low pH, nitrite is partially present in its undissociated form, nitrous acid ($HNO_2$, $pK_a = 3.29$). Under anoxic conditions, nitrous acid can disproportionate to nitrate and nitric oxide (equation 1) (41, 42), explaining the chemical

**TABLE 1** Prevalence of identified candidates for DNRA in aSRB[a]

| Name | Nitrate reduction | Nitrite reduction | NarK DEACI_1354 | NarG DEACI_1355 | DEACI_1836 | AsrA DEACI_4025 | AsrB DEACI_4026 | AsrC DEACI_4027 | Hcp DEACI_0275 | NosZ DEACI_0711 | NarG[1] | NapA[1] | NrfA[1] | NirB[1] | AsrA[1] | Hcp[1] | NosZ[1] | Reference |
|---|---|---|---|---|---|---|---|---|---|---|---|---|---|---|---|---|---|---|
| *Acididesulfobacillus acetoxydans*[T] | + | + | 100% | 100% | 100% | 100% | 100% | 100% | 100% | 100% | 52% | – | – | 34% | 43% | 43% | 34% | This study |
| *Thermodesulfobium narugense*[T] | + | + | 52% | 50% | 35% | – | – | – | 44% | – | 55% | – | – | 34% | – | 47% | – | (20) |
| *Thermodesulfobium acidiphilum*[T] | – | – | 52% | 49% | 36% | – | – | – | 43% | – | 56% | – | – | 33% | – | 46% | – | (21) |
| *Desulfosporosinus acidiphilus*[T] | + | + | – | – | 76% | 54% | 55% | 72% | 82% | – | – | – | – | 35% | 41% | 45% | – | (22) |
| *Desulfosporosinus acididurans*[T] | + | + | 82% | 77% | 74% | 56% | 54% | 73% | 81% | – | 52% | – | – | 37% | 44% | 43% | – | (23) |
| *Desulfosporosinus metallidurans*[T] | + | ND | – | 30% | 76% | – | – | – | 82% | – | – | – | – | 35% | – | 43% | – | (24) |
| *Desulfosporosinus* sp. I2 | + | – | – | 31% | 76% | – | – | 32% | 80% | – | – | – | 35% | 35% | – | 43% | – | (25) |
| *Desulfosporosinus* sp. BG | ND | ND | – | – | 77% | 55% | 53% | 72% | 81% | – | – | – | – | 36% | 43% | 42% | – | (26) |
| *Desulfosporosinus* sp. OT | ND | ND | – | – | 77% | 54% | 53% | 71% | 81% | – | – | – | – | 36% | 44% | 43% | – | (27, 28) |
| *Desulfosporosinus orientis*[T] | – | – | – | – | 75% | – | – | – | 81% | – | – | – | – | 36% | – | 44% | – | (29, 30) |
| *Desulfosporosinus hippei*[T] | – | ND | – | 30% | 76% | – | – | – | 82% | – | – | – | 36% | 35% | – | 43% | – | (31) |
| *Desulfosporosinus meridiei*[T] | – | ND | – | – | 76% | – | – | – | 81% | 46% | – | – | 37% | 34% | – | 43% | 37% | (30) |
| *Desulfosporosinus fructosivorans*[T] | – | ND | – | 30% | 73% | – | – | – | 79% | – | – | – | 35% | 34% | – | 43% | – | (32, 33) |
| *Desulfosporosinus lacus*[T] | – | – | – | – | 74% | – | – | – | 80% | – | – | – | 36% | 35% | – | 43% | – | (34) |
| *Desulfosporosinus youngiae*[T] | – | – | – | – | 76% | – | – | – | 81% | 43% | – | – | 36% | 35% | – | 43% | 36% | (35) |
| *Desulfitobacterium dehalogenans*[T] | + | – | – | – | 75% | – | – | 34% | 82% | 45% | – | 39% | 34% | 36% | – | 45% | 36% | (36) |
| *Desulfitobacterium hafniense*[T] | + | + | – | – | 76% | – | – | 31% | 82% | 43% | – | – | 34% | 36% | – | 44% | 35% | (37) |
| *Desulfitobacterium metallireducens*[T] | – | – | – | – | 78% | – | – | – | 80% | – | – | – | – | 33% | – | 44% | – | (38) |

*(Continued on next page)*

**TABLE 1** Prevalence of identified candidates for DNRA in aSRB[a] (*Continued*)

| Name | Nitrate reduction | Nitrite reduction | NarK DEACl_1354 | NarG DEACl_1355 | DEACl_1836 | AsrA DEACl_4025 | AsrB DEACl_4026 | AsrC DEACl_4027 | Hcp DEACl_0275 | NosZ DEACl_0711 | NarG[1] | Nap A[1] | NrfA[1] | NirB[1] | AsrA[1] | Hcp[1] | NosZ[2] | Reference |
|---|---|---|---|---|---|---|---|---|---|---|---|---|---|---|---|---|---|---|
| *Desulfitobacterium chlororespirans*[T] | – | ND | – | 67% | 75% | – | – | 31% | 82% | 43% | 53% | 39% | 33% | 34% | – | 44% | 35% | (39) |

[a]Of the closest genera to *Acididesulfobacillus*: *Desulfosporosinus* and *Desulfitobacterium*, the neutrophilic type strains are included (depicted in bold). Nitrate and nitrite reduction indicates the reported ability to reduce the compound. Hits with both a total score of >100 and identity percentage >30% are listed. For *Ds. meridiei*, the initial characterization paper did not observe nitrate reduction, whereas later on nitrate reduction was reported (24). Reference sequences were taken from [1]*Escherichia coli* strain K12 and [2]*Paracoccus denitrificans*.

formation of nitric oxide with the concomitant loss of nitrite in the 10 mM nitrate and 0.5 mM nitrite abiotic controls. In these chemical controls, no decrease in nitric oxide was observed nor a transformation to either ammonia or hydroxylamine. This excludes additional chemically formed intermediates from nitrite and nitric oxide in this experimental setup.

$$3\ HNO_2 \rightarrow NO_3^- + 2\ NO + H_2O + H^+ \tag{1}$$

Nitrite is produced by the nitrate reductase NarG; a known side activity of NarG is the reduction of nitrite to nitric oxide in the cytoplasm (43, 44). The percentage of produced nitric oxide through this mechanism differs significantly per organism as it is highly dependent on the extracellular and intracellular concentrations of nitrite, nitrate, and nitric oxide (45). While we did not measure nitrous oxide and dinitrogen gas, the fact that nitrogen balances in our experiments could be closed suggests that those are not major end products.

This study initially identified multiple candidates for the missing step for DNRA in *A. acetoxydans*, three located in the cytoplasm (AsrABC, Hcp, and DEACI_1836) and one in the periplasm (NosZ). None of these have been previously described as nitrite reductases. The trimeric anaerobic sulfite reductase AsrABC is an NADH-linked reductase induced by sulfite (46, 47). During sulfate reduction, *A. acetoxydans* reduces sulfite via the dissimilatory sulfite reductase complex DsrAB (DEACI_0668-0669) to DsrC-trisulfide (DEACI_0677), which is then reduced with the electrons received from the DsrMKJOP membrane complex (DEACI_0672-0676). Sulfate reduction takes place without the involvement of AsrABC, as confirmed by both proteome and transcriptome, suggesting either redundancy or another function for AsrABC. However, its upregulation during nitrate reduction points toward another function for AsrABC. Sulfite reductases have been linked to nitrite reduction, and it has been suggested that all sulfite reductases can reduce nitrite to ammonia (48). Interestingly, the *asrABC* deletion mutants of *Salmonella typhimurium* (43%, 40%, and 42% protein identity scores to *A. acetoxydans*, respectively) were still capable of nitrite reduction, whereas sulfite reduction halted (47). This indicates a role of this specific AsrAB in sulfite reduction, rather than nitrite metabolism; to complicate the interpretation of these results further, *S. typhimurium* encodes NrfA, a known nitrite-ammonifying enzyme (47). Heterologous expression of *S. typhimurium asrABC* in *E. coli* resulted in the ability of *E. coli* to reduce sulfite to sulfide (47). AsrABC has not been purified, and therefore, enzyme assays with either nitrite, nitric oxide, or hydroxylamine have not been performed; therefore, its physiological role, potential, and versatility remain enigmatic (49).

DEACI_1836 contains both the nitrite and sulfite reductase ferredoxin-like half domain (PF03460) and a nitrite and sulfite reductase 4Fe-4S domain (PF01077). These domains are characteristic of two types of nitrite reductases: the NADH-dependent NirB and the ferredoxin-dependent NirA (50, 51). Compared to NirB, DEACI_1836 lacks a flavine-adenine-dinucleotide prosthetic group (FAD) that facilitates electron transfer from NADH. Although DEACI_1836 shows higher homology to NirB than NirA (identity score <25%), both domains of NirA are conserved. An amino acid alignment of DEACI_1836 to described NirA sequences shows the conservation of the four cysteine residues that coordinate the four iron ions of the [4Fe4S] cluster and additionally coordinate the iron of the siroheme cofactor (File S5). The active site is preserved, which facilitates electron transfer from ferredoxin to the [4Fe4S] cluster (52, 53). In addition, next to DEACI_1836, a ferredoxin is encoded, which is highly upregulated in the transcriptome. Based on these observations, we hypothesize that DEACI_1836 reduces nitrite to ammonia with reduced ferredoxin as an electron donor. Reduced ferredoxin is generated during glycerol oxidation by decarboxylation of pyruvate to acetyl-CoA with the pyruvate ferredoxin oxidoreductase Pfor (DEACI_1130, DEACI_1144). Unsurprisingly, Pfor is highly abundant in both sulfate- and nitrate-reducing conditions. NirA from *P. aeruginosa* could reduce nitrite and sulfite in enzymatic assays using methyl viologen as an electron donor. Nitrite was reduced to ammonia in a 1:1 ratio, excluding nitric oxide

or hydroxylamine production. Additionally, NirA of *P. aeruginosa* could not use hydroxyl-amine as an electron acceptor in the presence of methyl viologen (54). Interestingly, the chloroplast NirA from *Spinacia oleracea* and *Cucurbita pepo* showed both nitrite and hydroxylamine reduction to ammonia coupled to methyl viologen oxidation although hydroxylamine was not considered the physiological electron acceptor (55, 56). NirB mutants of *S. typhimurium*, which encode both NarG and NapA, were used to test whether NirB could reduce nitrite to nitric oxide. No change was observed in nitric oxide production, ruling out nitric oxide production by NirB (43). This demonstrates that siroheme-containing nitrite reductases do not form nitric oxide from nitrite but could potentially reduce hydroxylamine besides its potential role in DNRA.

The hydroxylamine reductase Hcp was the fourth most abundant protein although no hydroxylamine was detected in any of the cultivations. Based on resting cell experiments, *A. acetoxydans* can reduce hydroxylamine to ammonia; however, the physiological role of Hcp and annotation as hydroxylamine reductase is controversial. Recent studies showed that although Hcp can reduce hydroxylamine, the actual physiological function is that of a high-affinity nitric oxide reductase (44, 57). As a nitric oxide reductase, Hcp can convert nitric oxide stoichiometrically into nitrous oxide under physiologically relevant conditions (57, 58). Likely, Hcp is a major contributing protein for handling nitrosative stress, as shown by the increased sensitivity to nitrosative stress of Hcp mutants of *S. typhimurium*, *Porphyromonas gingivalis*, and *Desulfovibrio gigas* (42%, 65%, and 48% protein identity scores to *A. acetoxydans*, respectively) (59–61). This is in line with the transcriptome of *Desulfovibrio desulfuricans,* which showed increased expression of *hcp* (69% protein identity score to *A. acetoxydans*) under nitrosative stress caused by supplemented nitric oxide (7.5 µM) under both sulfate- and nitrate-reducing conditions (62). In *A. acetoxydans,* nitric oxide could also be converted by the nitric oxide reductases NorBC. However, *norBC* was hardly expressed in the transcriptome and did not change expression level, potentially the expression of NorB was repressed by active DNRA (63). Probably, Hcp is the protein converting nitric oxide to nitrous oxide in the cytoplasm, preventing the accumulation of nitric oxide and minimizing nitrosative stress. This hypothesis is in line with the chemical data obtained in this study, which shows decreased nitric oxide accumulation in the nitrate-reducing cultures compared to the chemical control with nitrite. Furthermore, nitric oxide increases after metabolic activity halts. This aligns with the unexpected upregulation of the operon encoding the nitrous oxide reductase NosZ and ABC transporter NosDFY required for NosZ formation (64, 65). Increased nitrous oxide concentrations were shown to hamper DNRA upon oxic-to-anoxic transition in *Bacillus* sp. DNRA2, which could be alleviated by timely activated NosZ (43% protein identity score to *A. acetoxydans*) (66). In a similar fashion, the build-up of nitrous oxide from nitric oxide can also hamper DNRA activity in *A. acetoxydans*, explaining the role of NosZ during DNRA (66). Hence, the produced nitrous oxide is subsequently reduced to dinitrogen gas by NosZ with electrons supplemented from PetABC, although additional electron transfer pathways might be involved (67). The detected concentration of nitric oxide was in the micromolar range, whereas the other main conversion products including ammonia were in the millimolar range with closing nitrogen balances. Hence, this reductive nitric oxide flux seems to be minor in terms of nitrogen flux, yet important to maintain the viability of *A. acetoxydans* during nitrate reduction.

All combined, we propose a hypothetical pathway of dissimilatory nitrate reduction to ammonium in *A. acetoxydans* (Fig. 3). After the uptake of nitrate by NarK, nitrate is reduced to nitrite by NarG. From nitrite, the observed pathway splits into the ammoni-fication route and the nitric oxide route. Nitrite is likely reduced to ammonia by the previously undescribed nitrite reductase activity of the NADH utilizing AsrABC or by the ferredoxin-dependent DEACI_1836 or both. Produced ammonia ($pK_a$ = 9.25) is mainly present in its charged ammonium form at cytoplasmic pH, requiring active transport, which could be carried out by an Amt-type ammonium transporter. The nitric oxide route can start extracellularly or cytoplasmically. Extracellular nitrite disproportionates

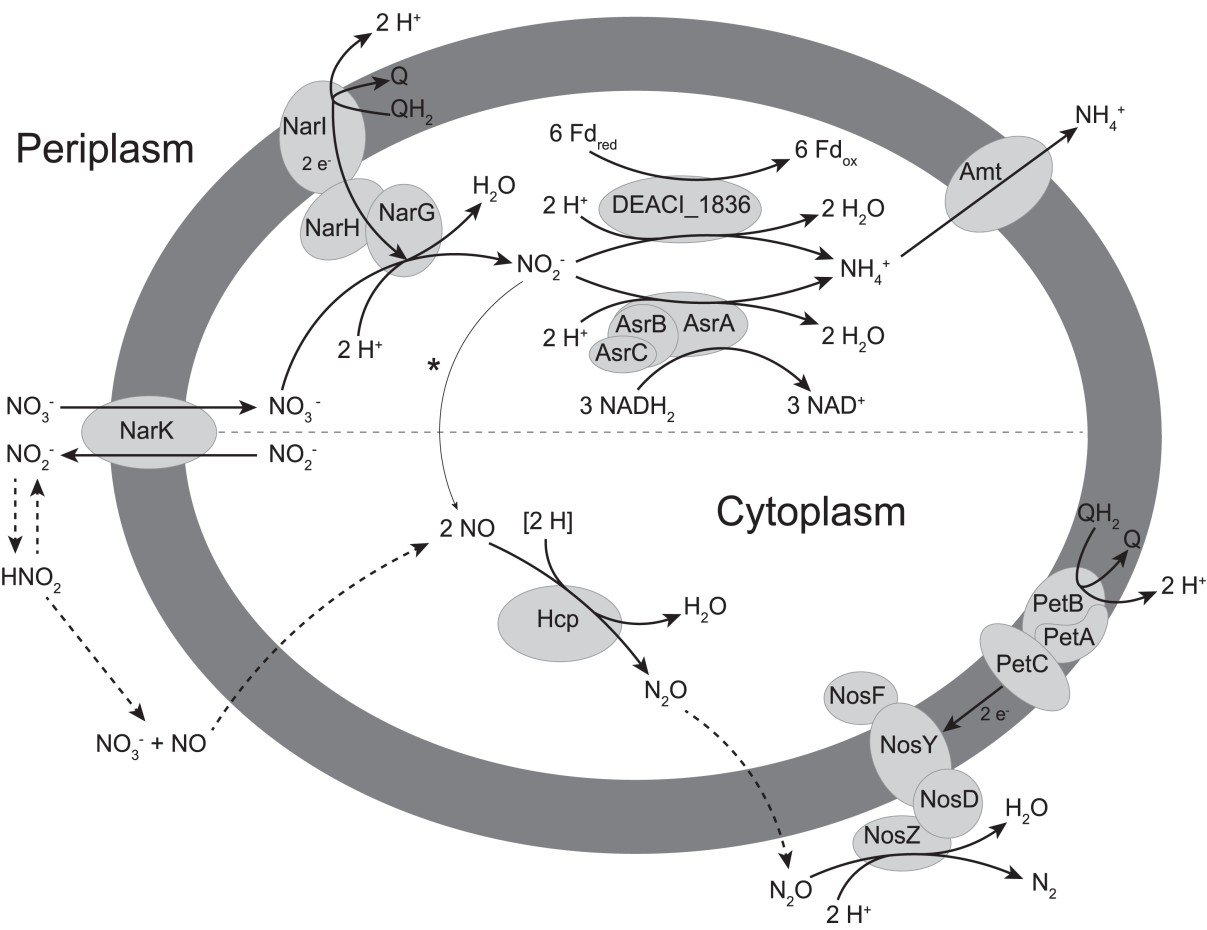

**FIG 3** Overview of the DNRA (top) and denitrification (bottom) pathways employed by *A. acetoxydans*. Nitrate is reduced to ammonium while accumulating nitrite and nitric oxide. The nitrogen balance closed completely, excluding the formation of other major end products or intermediates. The ferredoxin (DEACI_1837) potentially donating the electrons to DEACI_1836 can only transport a single electron as it contains a single [4Fe4S] cluster domain. The asterisk depicts the potential side nitrite reductase activity of NarG, and the dashed lines indicate abiotic steps.

into nitric oxide, which can freely permeate the membrane and enter the cell. Cytoplasmic nitrite can be converted (as side activity) by NarG into nitric oxide. Cytoplasmic nitric oxide can be reduced to nitrous oxide by Hcp, which is subsequently reduced to dinitrogen gas by NosZ (and NosFYD) with PetABC.

## Distribution of potential nitrite conversion pathways in sulfate-reducing bacteria

The ability to reduce nitrate is broadly distributed among the different phyla of SRB (62, 68, 69). SRB encoding a known nitrite reductase are more numerous than those encoding a known nitrate reductase, which is attributed to handling nitrosative stress caused by nitrite produced in their environmental vicinity (69, 70). However, the small number of studies on SRB examining either nitrite reduction or DNRA limits the understanding and prevalence of nitrite conversion pathways in aSRB. Based on the performed genome comparison, the combination of the projected DNRA and denitrification pathway is only present in *A. acetoxydans*. The reductive route from nitric oxide to dinitrogen gas is only conserved in a few SRB, none of which are acidophilic. Interestingly, it seems that DEACI_1836 is conserved in the majority of SRB although the similarity to sulfite reductases and lack of described NirA make it difficult to assign functional homology. Further study is required to understand the role of NirA, as currently described NirA are from phylogenetically very distinct organisms, which complicates

functional interpretation. In contrast, AsrABC is only conserved in *A. acetoxydans* and four acidophilic *Desulfosporosinus*. The common trait of these strains is the source of isolation, all were isolated from acid mine drainage environments. However, AsrA is also detected in neutrophils, such as *E. coli* and *S. typhimurium*; hence, this trait cannot be ascribed to being acidophilic or isolated from acid mine drainage environments. Aside from nitrite reduction, the most crucial step in tackling nitrosative stress is performed by Hcp, which was detected in all checked aSRB (44, 58, 71). Given the chemical disproportionation of nitrite into nitric oxide in anoxic acidic environments, perhaps Hcp is advantageous for handling nitrosative stress. The role of Hcp in anoxic acidic conditions is an interesting target for microbial physiology studies.

This study highlights the fact that DNRA is structurally and functionally more diverse than previously thought. These findings can help interpretation of (meta)genome data from various ecosystems on their DNRA potential and the nitrogen cycle.

## MATERIALS AND METHODS

### Strain used and cultivation conditions

*A. acetoxydans* DSM 29876[T] was retrieved from our own culture collection at the Laboratory of Microbiology (Wageningen University & Research, Wageningen, the Netherlands). All *A. acetoxydans* cultures were statically cultivated at 30°C using 217 mL serum vials containing 100 mL anoxic medium at pH 5.0. The medium is a modification of the medium used by Stams et al. (72) and consisted of (g/L): $Na_2HPO_4 \cdot 2H_2O$, 0.53; $KH_2PO_4$, 0.41; $NH_4Cl$, 0.3; $NaCl$, 0.3; $MgCl_2 \cdot 6H_2O$, 0.1; and $CaCl_2 \cdot H_2O$, 0.11. As a redox indicator, 0.5 mg/L resazurin was added to the medium. Furthermore, acid and alkaline trace element solutions (1 mL/L) and a vitamin solution (0.2 mL/L) were added. The acid trace element solution consisted of (mM) $H_3BO_4$, 1; $CoCl_2$, 0.5; $CuCl_2$, 0.1; $FeCl_2$, 7.5; $NiCl_2$, 0.1; $MnCl_2$, 0.5; $ZnCl_2$, 0.5; and HCl, 50. The alkaline trace element solution consisted of (mM) $Na_2MoO_4$, 0.1; $Na_2SeO_3$, 0.1; $Na_2WO_4$, 0.1; and NaOH, 50. The vitamin solution consisted of (g/L) biotin, 0.02; cyanocobalamin, 0.1; niacin, 0.2; *p*-aminobenzoic acid, 0.1; pantothenic acid, 0.1; pyridoxine, 0.5; riboflavin, 0.1; and thiamine, 0.2. Chemicals were obtained from Sigma-Aldrich (Merck KGaA, Darmstadt, Germany) unless otherwise stated. Medium was supplemented with 5 mM glycerol, 0.1 g/L yeast extract, 1.5 mM of $Na_2S$ as a reducing agent, and either 20 mM $NaNO_3$ (nitrate) or 20 mM $NaSO_4$ (sulfate) as an electron acceptor. The headspace was filled with $N_2/CO_2$ (1.5 atm, 80:20, vol/vol) as a gas phase. All compounds were autoclaved except the vitamins, which were filter-sterilized through a 0.22-µM pore size polyethersulfone filter (Advanced Microdevices, Tepla, India). *A. acetoxydans* was transferred at least 10 times to ensure complete adaptation to either nitrate or sulfate as a sole electron acceptor. To obtain growth curves and transcriptomic and proteomic data, a 1% (vol/vol) inoculum was used, and experiments were performed in quadruplicate unless stated otherwise.

### Resting cell experiments

For resting cell experiments, 100 mL of cell culture was harvested from cultivations in mid-exponential phase in an anaerobic tent (Coy, Grass Lake, MI, USA). The cell culture was centrifuged at $6,000 \times g$ for 10 min (ThermoFisher Scientific, Waltham, MA, USA) and washed with mineral salt medium (MSM) not containing nitrate, glycerol, yeast extract, or vitamins. The pellet was resuspended in this MSM to 10% of the original volume and transferred to sterile 27 mL Hungate tubes, in which the gas phase was exchanged with $N_2/CO_2$ (80:20, vol/vol). To each tube, 2 mM glycerol was added as an electron donor, and to start the experiment, either $NaNO_3$ (1 mM), $NaNO_2$ (0.5 mM), or $NH_2OH \cdot HCl$ (hydroxylamine hydrochloride, 0.5 mM) were added as an electron acceptor.

## Analytical methods

Liquid and gas samples were taken from cultures growing under sulfate- and nitrate-reducing conditions and from the resting cell experiments. Liquid samples were taken to determine pH, optical density at 600 nm ($OD_{600}$), and the concentrations of glycerol, organic acids, sulfate, nitrate, nitrite, hydroxylamine, and ammonium. Growth was monitored by $OD_{600}$ using a spectrophotometer UV-1800 spectrophotometer (Shimadzu, Kyoto, Japan). Glycerol and organic acids were determined with high-performance liquid chromatography (HPLC) using a Shimadzu LC2030c plus (Shimadzu) with a differential refractive index detector Shimadzu RID-20A (Shimadzu) and equipped with a Shodex SH1821 column (Shodex, Kyoto, Japan) operated at 45°C. As eluent, 5 mM sulfuric acid was used at a flow rate of 0.8 mL/min. Sulfate and nitrate were measured by anion exchange chromatography (IC) on a Dionex ICS-2100 (Dionex, Sunnyvale, CA, USA) equipped with a Dionex IonPac AS19 column (Dionex) operated at 30°C. As eluent, KOH 22% (wt/vol) was used in a gradient ranging from 10 to 40 mM at a flow rate of 0.4 mL/min. For both HPLC and IC, the lower detection limit of measured compounds was 100 µM. Nitrite was measured using the Griess assay. In technical triplicates, 100 µL of supernatant was transferred to a 96-well plate containing 100 µL of Griess reagent, consisting of 50 µL 1% (wt/vol) sulfanilic acid in 1 M HCl and 50 µL of 0.1% (wt/vol) naphthylethylene diamine dihydrochloride. After 10 min incubation at room temperature, the absorbance was measured at 540 nm with a SpectraMax 190 microplate reader (Molecular Devices, San Jose, CA, USA). Hydroxylamine was measured using a colorimetric hydroxylamine assay (73). To a 2 mL Eppendorf (Eppendorf, Hamburg, Germany), 20 µL 50 mM phosphate buffer (pH 7), 160 µL demineralized $H_2O$, 200 µL sample, 40 µL 12% (wt/wt) trichloroacetic acid, 200 µL 1% (wt/wt) 8-hydroxyquinoline, and 200 µL 1 M $Na_2CO_3$ were added. The solution was incubated at 100°C for 1 min and afterward incubated at room temperature for 15 min. In technical duplicates, the absorbance was measured at 705 nm. Ammonium was measured using the colorimetric Spectroquant Ammonium Test (Merck KGaA, Darmstadt, Germany). The absorbance was measured at 690 nm in technical duplicates. The lower detection limits for nitrite, hydroxylamine, and ammonium were 2.5, 10, and 100 µM, respectively. Gas samples were taken to determine nitric oxide using a NOA 280i nitric oxide analyzer (GE Healthcare, Chicago, IL, USA). For all gas analyses, 100 µL gas samples were injected in the nitric oxide analyzer with a suction rate of 11.6 mL/min. The lower detection limit of nitric oxide was ~0.5 parts per million (ppm).

## RNA extractions and sequencing

For transcriptomics, 100 mL of cell culture was harvested from cultivations in mid-exponential phase in an anaerobic tent (Coy). The cell culture was added to 200 mL of ice-cold sterile reduced medium, followed by centrifugation at 10,000 × $g$ for 10 min at 4°C (ThermoFisher Scientific). After centrifugation, the pellet was resuspended in 10 mL tris-EDTA (TE) buffer, transferred to sterile 50 mL tubes (Greiner, Frickenhausen, Germany), and centrifuged at 10,000 × $g$ for 10 min at 4°C. The obtained pellet was snap-frozen in liquid nitrogen and stored at −80°C. For RNA isolation, the cells were lysed, and the proteins precipitated by using the MasterPure Gram Positive DNA Purification kit (Epicentre, Madison, WI, USA), without the RNAse step. β-mercaptoethanol was added to the lysis solution. RNA was extracted by using the Maxwell 16 LEV simplyRNA Purification kit on the Maxwell 16 platform (Promega, Fitchburg, WI, USA). Samples were sent to Novogene (Novogene, Cambridge, UK) for quality control. At Novogene, rRNA depletion was performed using the Illumina Ribo-Zero Plus rRNA Depletion Kit (Illumina, San Diego, CA, USA) and sequenced using the NovaSeq6000 (Illumina) resulting in paired-end reads of 150 bp.

## Transcriptomics data analysis

Raw sequence data were trimmed for Illumina adapters and paired by using Trimmo-matic 0.38 (74) with the following settings: SLIDINGWINDOW: four bases, quality ≥20, average read score >5, MINLEN: 100. Quality trimmed paired-end reads were checked by using FastQC 0.11.9 and MultiQC 1.11 (75, 76). The reads were mapped against the reference genome of *A. acetoxydans* (NCBI accession number: SAMEA6497484) using BWA-MEM 0.7.17 (77). Before mapping, 66 structural RNA (rRNA, sRNA, and tRNA) genes were removed from the data set. Count matrices with mapped reads were obtained by using featureCounts 2.0.1 with paired-end reads enabled (78). Mapping quality was assessed with SAMtools (flagstat) 2.0.4 (79). Mapped reads were converted into TPM to compare expression levels within samples. To test for significant statistical differences in gene expression between groups, we analyzed the read counts with the R package DESeq2 1.30.1 (80) using R 4.2.1. Heatmaps of both transcriptome and proteome were generated with the R package pheatmap and for visualization scaled per row. Differentially expressed genes were identified as genes with an expression change of at least fourfold and an adjusted *P* value of ≤0.01.

## Protein extractions

For proteomics, 100 mL of cell culture was harvested of cultivations in mid-exponential phase in an anaerobic tent (Coy). Cultures were harvested by centrifuging at 10,000 × *g* for 10 min at 4°C. Pellets were washed twice and resuspended in 1 mL of 100 mM Tris-HCL (pH 8). Solution was sonicated using a bandelin sonicator (Bandelin, Berlin, Germany) and a MS72 microtip probe (Bandelin, Berlin, Germany) sonicating at 40% amplitude for 5 min with cycles of 3 s pulse and 3 s pause while cooling on ice. Cell debris was removed by centrifugation at 10,000 × *g* at 4°C, and the protein concentration of the supernatant was determined using the Pierce BCA Protein Assay Kit (ThermoFisher Scientific) using a bovine serum albumin standard. The supernatant was reduced with 15 mM dithiothreitol and incubated at 45°C for 30 min. Proteins were unfolded with 6 M urea and alkylated in 20 mM acrylamide and incubated at 21°C for 30 min. The pH was adjusted to 7 using 10% trifluoroacetic acid (TFA). Protein aggregation capture was used to capture proteins by adding 1-μm-diameter magnetic carboxylate-modified beads (GE Healthcare) (81, 82). Protein aggregation was induced by the addition of 2.5 times the working volume of acetonitrile followed by 20 min shaking at room temperature. The solution was allowed to settle for 5 min and, during this and the consequent steps, beads were retained using a magnet, and after 60 s, the supernatant was removed. Beads were washed in sequence with 70% ethanol and acetonitrile. Proteins were digested by adding 0.5 μg sequencing grade trypsin in 100 μL of 50 mM ammonium bicarbonate and incubated overnight at room temperature. Peptides were acidified to pH 3 using 10% TFA. Afterward, samples were pulse-centrifuged and the supernatant filtered through a double-layered C8 solid-phase extraction disk (CDS Analytical, Oxford, PA, USA). Trapped peptides in the extraction disk were eluted with a 50% acetonitrile and 0.1% formic acid solution. Peptides were concentrated using a Concentrator plus (Eppendorf, Hamburg, Germany) to evaporate the organic solvent.

## Proteomics

Per sample, 0.5 μg of protein was loaded directly onto a 0.10- × 250-mm ReproSil-Pur 120 C18-AQ 1.9-μm beads analytical column (beads prepared in-house, column from Dr. Maisch, Germany) at a constant pressure of 82.5 MPa (flow rate of circa 600 nL/min) with 1% formic acid in water and eluted at a flow of 0.5 μL/min with a 50 min linear gradient from 9% to 34% acetonitrile in water and 1% formic acid with a Thermo EASY nanoLC1000 (ThermoFisher Scientific). An electrospray potential of 3.5 kV was applied directly to the eluent via a stainless-steel needle fitted into the waste line of a micro cross that was connected between the nLC and the analytical column. On the connected Orbitrap Exploris 480 (ThermoFisher Scientific), mass spectrometry (MS) and

tandem mass spectrometry (MSMS) automatic gain control targets were set to 300% and 100%, respectively, or maximum ion injection times of 50 ms (MS) and 30 ms (MSMS) were used. Higher energy collisional dissociation (HCD) fragmented (isolation width 1.2 $m/z$ and 28% normalized collision energy) MSMS scans in a cycle time of 1.1 recording the most abundant 2–5+ charged peaks in the MS scan in a data-dependent mode (Resolution 15000, threshold 2e4, 15 s exclusion duration for the selected $m/z$ ± 10 ppm). Liquid chromatography tandem mass spectrometry (LC-MSMS) runs with all MS/MS spectra obtained were analyzed with MaxQuant 1.6.3.4 (83). Peptides and proteins were considered reliable for further analysis when they had a false discovery rate <1% and when at least two peptides of a protein were identified of which at least one was unique and one unmodified. Data analysis was performed using Perseus version 1.6.2.1 (84). The relative protein abundances were represented as log10-transformed LFQ values. Student's $t$ test was performed using the "LFQ intensity" columns obtained with a significance level of $P \leq 0.05$ and S0 = 0.1. The nano-liquid chromatography tandem mass spectrometry system and data quality were checked with PTXQC using the MaxQuant result files (85).

## ACKNOWLEDGMENTS

We thank Ida Peterse for the technical support in nitrite and nitric oxide measurements. We thank Arjan Pol for his advice on the nitrogen metabolism and suggestions for potentially formed intermediates.

This work was financed by the Soehngen Institute for Anaerobic Microbiology Gravitation Program (SIAM 024.002.002), a Gravitation grant of the Dutch Ministry of Education, Culture and Science.

## AUTHOR AFFILIATIONS

[1]Laboratory of Microbiology, Wageningen University & Research, Wageningen, The Netherlands
[2]Microcosm Earth Centre, Philipps-Universität Marburg & Max Planck Institute for Terrestrial Microbiology, Marburg, Germany
[3]Laboratory of Biochemistry, Wageningen University & Research, Wageningen, The Netherlands
[4]Centre for Living Technologies, Alliance TU/e, WUR, UU, UMC Utrecht, Utrecht, The Netherlands
[5]Department of Microbiology, Radboud Institute for Biological and Environmental Sciences, Radboud University, Nijmegen, The Netherlands
[6]Department of Environmental Sciences for Sustainability, IE University, Segovia, Spain

## AUTHOR ORCIDs

Reinier A. Egas  http://orcid.org/0000-0003-1823-2179
Julia M. Kurth  http://orcid.org/0000-0002-1221-1230
Diana Z. Sousa  http://orcid.org/0000-0003-3569-1545
Cornelia U. Welte  http://orcid.org/0000-0002-1568-8878
Irene Sánchez-Andrea  http://orcid.org/0000-0001-6977-3026

## FUNDING

| Funder | Grant(s) | Author(s) |
| --- | --- | --- |
| Soehngen Institute of Anaerobic Microbiology (SIAM) | 024.002.002 | Reinier A. Egas |

## AUTHOR CONTRIBUTIONS

Reinier A. Egas, Conceptualization, Formal analysis, Investigation, Visualization, Writing – original draft, Writing – review and editing | Julia M. Kurth, Investigation, Writing –

review and editing | Sjef Boeren, Formal analysis, Methodology, Resources | Diana Z. Sousa, Conceptualization, Supervision, Writing – review and editing | Cornelia U. Welte, Conceptualization, Funding acquisition, Investigation, Supervision, Validation, Writing – original draft, Writing – review and editing | Irene Sánchez-Andrea, Conceptualization, Funding acquisition, Investigation, Supervision, Writing – original draft, Writing – review and editing

## DATA AVAILABILITY

The raw RNA-Seq data sets have been deposited to the European Nucleotide Archive: https://www.ebi.ac.uk/ena, under accession number PRJEB64595. The mass spectrometry proteomics data have been deposited to the ProteomeXchange Consortium via the PRIDE partner repository: https://www.ebi.ac.uk/pride, with the data set identifier PXD045135 (86).

## ADDITIONAL FILES

The following material is available online.

### Supplemental Material

**File S1 (mSystems00967-23-s0001.docx).** Growth of *Acididesulfobacillus acetoxydans* with nitrate and glycerol.
**File S2 (mSystems00967-23-s0002.docx).** Resting cell experiments of *Acididesulfobacillus acetoxydans*.
**File S3 (mSystems00967-23-s0003.xlsx).** Transcriptome and proteome data.
**File S4 (mSystems00967-23-s0004.docx).** Upregulated molybdopterin containing oxidoreductases.
**File S5 (mSystems00967-23-s0005.docx).** Multiple sequence alignment of NirA and DEACI_1836.

### Open Peer Review

**PEER REVIEW HISTORY (review-history.pdf).** An accounting of the reviewer comments and feedback.

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
