## [Reviewer comments · mSystems]

A novel mechanism for dissimilatory nitrate reduction to ammonium in *Acididesulfobacillus acetoxydans*

Reinier Egas, Julia Kurth, Sjeff Boeren, Diana Sousa, Cornelia Welte, and Irene Sánchez-Andrea

Corresponding Author(s): Irene Sánchez-Andrea, IE University

Review Timeline:

Submission Date:	September 9, 2023
Editorial Decision:	October 10, 2023
Revision Received:	December 5, 2023
Accepted:	December 25, 2023

Editor: Liyuan Ma

Reviewer(s): Disclosure of reviewer identity is with reference to reviewer comments included in decision letter(s). The following individuals involved in review of your submission have agreed to reveal their identity: Sukhwan Yoon (Reviewer #2); Zhe-Xue Quan (Reviewer #3)

Transaction Report:

DOI: <https://doi.org/10.1128/msystems.00967-23>

October 10, 2023

Dr. Irene Sánchez-Andrea
Laboratory of Microbiology, Wageningen University, Stippeneng 4, 6708 WE, Wageningen, The Netherlands
Stippeneng 4
Wageningen 6708 WE
Netherlands

Re: mSystems00967-23 (A novel mechanism for dissimilatory nitrate reduction to ammonium in *Acididesulfobacillus acetoxydans*)

Dear Dr. Irene Sánchez-Andrea:

Thank you for submitting your manuscript to mSystems. We have completed our review and I am pleased to inform you that, in principle, we expect to accept it for publication in mSystems. However, acceptance will not be final until you have adequately addressed the reviewer comments.

Preparing Revision Guidelines

Please return the manuscript within 60 days; if you cannot complete the modification within this time period, please contact me. If you do not wish to modify the manuscript and prefer to submit it to another journal, please notify me of your decision immediately so that the manuscript may be formally withdrawn from consideration by mSystems.

Sincerely,

Liyuan Ma

Editor, mSystems

Journals Department
Reviewer comments:

Reviewer #1 (Comments for the Author):

For the manuscript, "A novel mechanisms for dissimilatory nitrate reduction to ammonium in *Acidithiobacillus acetoxidans*", the authors attempt to elucidate the mechanisms whereby this acidophilic sulfate reducing bacterium can complete the transformation of nitrate to ammonium without an annotated nitrite reductase. The authors do an excellent job of confirming that *A. acetoxidans* does do DNRA and of following intermediates of nitrate reduction to ammonium in resting cell experiments. I really liked the transcriptome and proteome work by the authors. The conclusions were well-reasoned and at the same time not too presumptuous. *A. acetoxidans* may use an annotated sulfite reductase that may actually be a NirA nitrite reductase (DEACI_1836) or a sulfite reductase AsrABC or both for nitrite reduction to ammonium. Nitrite and sulfite reductases are notoriously promiscuous, so these results are not surprising and yet, the authors actually have some evidence for their implication in nitrite reduction. I also like the additional interpretations around the Hep and hydroxylamine reduction and nitrous oxide reduction for nitrosative stress relief in this organism. The authors did a good job of giving a bigger picture of the whole nitrate reduction to ammonium pathways, not just the nitrite reductase portion. I think this manuscript sets the stage well for gene knock experiments to confirm the nitrite reductase hypotheses and also has implications for interpretation of metagenomic sequencing data from extreme environments. There are a few places in this manuscript that could benefit from some minor revisions.

Minor revision recommendations

- Line 89: missing word, nitrite as a free ...
- Line 90: missing word, as an intermediate...
- Line 101: missing NasB. In line 87 the authors specifically mention NasB as a possible assimilatory nitrite reductase, but don't add it to Table 1. The genome of *A. acetoxidans* does not encode an annotated NasB. The assimilatory nitrite reductases should be mentioned not just the dissimilatory ones.
- Figure 1: I had trouble distinguishing the colors here. The nitrite and nitric oxide lines are indistinguishable.
- Line 146: Supplemental figure S2, graph B. It seems to be missing the nitrite line in the graph.
- Line 160: Its super interesting that the sulfate reduction pathways are constitutively expressed. This is another valuable contribution of this manuscript to understanding the physiology of these acidophilic microbes.
- Figure 2: I don't understand the figure color explanation. Yellow is 2 and purple is -2. How does this relate to the legend text (line 848) of normalized counts and LFQ values?
- Line 186: I like this paragraph of description for DEACI_1836. It fits well with the discussion (Line 328)
- Line 205: word choice, I am not clear what did not change expression levels.
- Line 261: According the Table 1, the authors show that *T. narugense* does have a NirB homolog. Could this assimilatory nitrite reductase be used for ammonification?
- Line 265: According to Table 1, *D. hafniense* does have a Hcp, but does not have DEACI_1836 homolog.
- Table 1:
Line 863: Add *Thermodesulfobium* as species mentioned in the legend text
I noticed one mistake, *D. hafniense* strain DCB-2 T does encode a NosZ protein (also line 267). Perhaps it is less than 30% similar to *Paracoccus*, which is why it wasn't included here. Its worth checking.
- Line 300: I think this is well-reasoned. Nitrous oxide production, if any, was probably very low and was likely converted to nitrogen gas quickly.
- Line 319: Again, I like how the authors leave this as a very cautious hypothesis.
- Line 329: missing word, to a described NirA
- Line 332: I like that the authors show this alignment. It is rough, but key active site domains are conserved.
- Line 338: word choice, please clarify what you mean by "both conditions"
- Line 356: missing word, As a nitric oxide...
- Line 358: word choice, "factor"
- Line 396: word choice, encoding a known nitrite reductase...
- Line 406: what do the authors mean by "widespread"?
- Line 409: According to Table 1, AsrC is detected in neutrophiles, not AsrA.
- Line 412: Hcp is detected in all examined (a)SRB and neutrophilic SRB as well. I'm kind of getting mixed up in this paragraph about the comparisons between (a)SRB and all SRB. Given the next sentence about using Hcp as a marker, perhaps the authors want to stay focused just on the (a) SRB.

- Line 425: missing word, The medium is a modification...
- Line 481: missing comma, In technical duplicates, the absorbance...
- Line 576: I could not access the RNA-seq data using this accession number or the organism name.
- Line 578: I could not access the proteomics data using this this dataset identifier or the organism name.

Reviewer #2 (Comments for the Author):

The authors report their recent discovery of a DNRA phenotype in a rather unexpected acidophilic bacterium without *nrfA* or *nirB*, the previously known nitrite reductase genes, in its genome. The manuscript is very well-written in general, and the experimental designs and procedures and the results all appear sound. I would really appreciate the authors' adherence to strong scientific fundamentals in preparing their manuscript, which made the review process a very pleasant process. I have only two major concerns: 1) the authors' negligence to the recent publications in ASM journals that would provide very strong supports to the findings reported in this current study (for example, Heo et al., 2020 AEM paper and Yoon et al., 2023 mBio paper), and 2) heavy reliance on the negative proteomics data for hypothesizing the enzymes pertaining to DNRA metabolism. Proteomics, although immensely popular today, has many drawbacks: that the results can never be discussed quantitatively, and that membrane proteins cannot be reliably detected. I have several additional minor comments that may help the authors improve the quality of their manuscript.

Line 188 - 193: I would only discuss about the two domains which have predicted functions. A 'domain of unknown function' carries no important information.

Line 199 - 200: The Yoon et al., 2023 mBio paper may help explain the presence of *nosZ* in the genome and the purpose of its expression to the microorganism.

Line 227 -228: This is odd. Possibly DEACI_3264 may be a structural protein.

Line 208 -232: This entire section is a discussion of why some of the genes possibly involved in N cycling are not likely as candidates for DNRA or nitric oxide reduction. Most of these are reiterated in the discussion section. I would recommend summarizing this section into a supplementary table and remove it.

Line 247 - 268: This section belongs to the discussion section. As this manuscript is a bit too long for what it contains, I would recommend combining the result and discussion sections. That would remove redundancies and help the readers' focus on the key findings.

Line 289: The sentence that starts with "In this chemical controls..." is a bold statement, unless NO formation is stoichiometric to NO₂⁻ loss. NO is a very difficult substance to hold stable, even just with pure water and a very small amount of oxygen penetrating into the system.

Line 313: Whenever the authors bring up the previous studies on the orthologues of the genes / enzymes in other organisms, they should mention about the similarity of the enzymes in those organisms from that in *Acididesulfobacillus acetoxydans*.

Line 365: This is one of the examples, where referring to the Heo et al., 2021 paper would be very helpful.

Line 371: This statement is an over-simplification. *NosZ* can receive electrons from many different electron donor compounds via varying electron transfer pathways.

Line 381: Einsle et al., 2011 Methods in Enzymology paper indicates that NO may be intermediates of *NrfA*, and Heo et al., 2021 paper and Yoon et al., 2023 paper support the possibility with physiological data. Although NO does not get detected extracellularly with the analytical equipment used by the authors, it is highly likely that NO is one of the intermediates of this enigmatic DNRA pathway.

Line 388: This hypothesis is too much of a stretch. It is highly unlikely that nitrite can be converted to NO via *NarG* activity.

Line 401: By definition, 'Denitrification' is reduction of dissolved nitrogen species (NO₃⁻ and/or NO₂⁻) to gaseous species (NO, N₂O, and N₂). I would be very careful to term NO-to-N₂O reduction as 'partial denitrification'.

Line 402-404: This sentence needs to be rewritten with better clarity.

Line 409-410: If *AsrA* is a periplasmic or cytoplasmic enzyme, the environmental pH should not be an important factor in its

possession, expression, or activity.

Line 413-415: I strongly disagree with this statement. Hcp is too wide-spread to be used as a marker gene for DNRA.

Line 431: Why were alkaline trace elements used for an acidic medium?

Line 436: 0.1 g/L yeast extract may be a serious problem here. The nitrogen content of a typical yeast extract powder is around 10%, and a substantial portion of this nitrogen may be mineralized to NH_4^+ . Perhaps, this was why the authors supplemented their results with the resting cell experiments. The authors should mention this in the methods or result section.

Line 488: I doubt that concentration of dissolved NO was measured. The detection limit should be presented in terms of the gaseous concentration.

Line 502-505: It appears that some important details are missing for the RNASeq procedure. It is often unnecessary to list all the details for extraction, processing, and sequencing, but considering all the minor details presented, some more detail would be needed here.

Line 516: This unit TPM is not normalized by the gene length. Large genes will have higher TPM values.

Reviewer #3 (Comments for the Author):

In this study, authors analyzed the pathway of dissimilatory nitrate reduction to ammonium (DNRA) in *Acidithiobacillus acetoxidans* physiology test. With comparative analysis of transcriptomics and proteomics, authors found that the NADH-linked sulfite reductase AsrABC and/or a putatively ferredoxin-dependent homolog of the nitrite reductase NirA (DEACI_1836) are the key enzyme(s) of nitrite reduction to ammonia. This work extended the knowledge of DNRA especially at extreme environments.

The topic is interesting, the conclusion is supported with solid results. The manuscript is well written.

Minor comments:

1) In Fig.1, I think what you determined is ammonium not ammonia. Please carefully differentiate the both.

2) In Supplementary Table 1, "NO-" should be changed to "NO₂-"; "glycerol" and "acetate" should be changed to "glycerol [mM]" and "acetate [mM]", respectively. Furthermore, the data of glycerol is not matched with Fig. S1, please check it.

9-30-23

For the manuscript, “A novel mechanisms for dissimilatory nitrate reduction to ammonium in *Acidithiobacillus acetoxidans*”, the authors attempt to elucidate the mechanisms whereby this acidophilic sulfate reducing bacterium can complete the transformation of nitrate to ammonium without an annotated nitrite reductase. The authors do an excellent job of confirming that *A. acetoxidans* does do DNRA and of following intermediates of nitrate reduction to ammonium in resting cell experiments. I really liked the transcriptome and proteome work by the authors. The conclusions were well-reasoned and at the same time not too presumptuous. *A. acetoxidans* may use an annotated sulfite reductase that may actually be a NirA nitrite reductase (DEACI_1836) or a sulfite reductase AsrABC or both for nitrite reduction to ammonium. Nitrite and sulfite reductases are notoriously promiscuous, so these results are not surprising and yet, the authors actually have some evidence for their implication in nitrite reduction. I also like the additional interpretations around the Hep and hydroxylamine reduction and nitrous oxide reduction for nitrosative stress relief in this organism. The authors did a good job of giving a bigger picture of the whole nitrate reduction to ammonium pathways, not just the nitrite reductase portion. I think this manuscript sets the stage well for gene knock experiments to confirm the nitrite reductase hypotheses and also has implications for interpretation of metagenomic sequencing data from extreme environments. There are a few places in this manuscript that could benefit from some minor revisions.

Minor revision recommendations

- Line 89: missing word, nitrite as *a* free ...
- Line 90: missing word, as *an* intermediate...
- Line 101: missing NasB. In line 87 the authors specifically mention NasB as a possible assimilatory nitrite reductase, but don't add it to Table 1. The genome of *A. acetoxidans* does not encode an annotated NasB. The assimilatory nitrite reductases should be mentioned not just the dissimilatory ones.
- Figure 1: I had trouble distinguishing the colors here. The nitrite and nitric oxide lines are indistinguishable.
- Line 146: Supplemental figure S2, graph B. It seems to be missing the nitrite line in the graph.
- Line 160: Its super interesting that the sulfate reduction pathways are constitutively expressed. This is another valuable contribution of this manuscript to understanding the physiology of these acidophilic microbes.
- Figure 2: I don't understand the figure color explanation. Yellow is 2 and purple is -2. How does this relate to the legend text (line 848) of normalized counts and LFQ values?
- Line 186: I like this paragraph of description for DEACI_1836. It fits well with the discussion (Line 328)
- Line 205: word choice, I am not clear what did not change expression levels.
- Line 261: According the Table 1, the authors show that *T. narugense* does have a NirB homolog. Could this assimilatory nitrite reductase be used for ammonification?

- Line 265: According to Table 1, *D. hafniense* does have a Hcp, but does not have DEACI_1836 homolog.
- Table 1:
Line 863: Add *Thermodesulfobium* as species mentioned in the legend text
I noticed one mistake, *D. hafniense* strain DCB-2^T does encode a NosZ protein (also line 267). Perhaps it is less than 30% similar to *Paracoccus*, which is why it wasn't included here. Its worth checking.
- Line 300: I think this is well-reasoned. Nitrous oxide production, if any, was probably very low and was likely converted to nitrogen gas quickly.
- Line 319: Again, I like how the authors leave this as a very cautious hypothesis.
- Line 329: missing word, to *a* described NirA
- Line 332: I like that the authors show this alignment. It is rough, but key active site domains are conserved.
- Line 338: word choice, please clarify what you mean by "both conditions"
- Line 356: missing word, As *a* nitric oxide...
- Line 358: word choice, "factor"
- Line 396: word choice, encoding a *known* nitrite reductase...
- Line 406: what do the authors mean by "widespread"?
- Line 409: According to Table 1, AsrC is detected in neutrophiles, not AsrA.
- Line 412: Hcp is detected in all examined (a)SRB and neutrophilic SRB as well. I'm kind of getting mixed up in this paragraph about the comparisons between (a)SRB and all SRB. Given the next sentence about using Hcp as a marker, perhaps the authors want to stay focused just on the (a) SRB.
- Line 425: missing word, *The* medium is a modification...
- Line 481: missing comma, In technical duplicates, the absorbance...
- Line 576: I could not access the RNA-seq data using this accession number or the organism name.
- Line 578: I could not access the proteomics data using this this dataset identifier or the organism name.

Point-by-point responses to received reviewer comments

We thank the reviewers for critically reading our manuscript and for their constructive comments and suggestions to improve it. We highly appreciate the time and rigor invested in their suggested revisions and the points addressed. We would also like to thank the reviewers' enthusiasm for our work.

Below, we have provided a point-by-point response (blue) to the referee's comments. The made adjustments are incorporated in the revised manuscript. For readability we added quotations to this file, the Marked-Up Manuscript contains all of the made changes. Thanks again for your contribution to the improvement of this manuscript.

Reviewer #1

For the manuscript, "A novel mechanisms for dissimilatory nitrate reduction to ammonium in *Acididesulfobacillus acetoxydans*", the authors attempt to elucidate the mechanisms whereby this acidophilic sulfate reducing bacterium can complete the transformation of nitrate to ammonium without an annotated nitrite reductase. The authors do an excellent job of confirming that *A. acetoxydans* does do DNRA and of following intermediates of nitrate reduction to ammonium in resting cell experiments. I really liked the transcriptome and proteome work by the authors. The conclusions were well-reasoned and at the same time not too presumptuous. *A. acetoxygens* may use an annotated sulfite reductase that may actually be a NirA nitrite reductase (DEACI_1836) or a sulfite reductase AsrABC or both for nitrite reduction to ammonium. Nitrite and sulfite reductases are notoriously promiscuous, so these results are not surprising and yet, the authors actually have some evidence for their implication in nitrite reduction. I also like the additional interpretations around the Hep and hydroxylamine reduction and nitrous oxide reduction for nitrosative stress relief in this organism. The authors did a good job of giving a bigger picture of the whole nitrate reduction to ammonium pathways, not just the nitrite reductase portion. I think this manuscript sets the stage well for gene knock experiments to confirm the nitrite reductase hypotheses and also has implications for interpretation of metagenomic sequencing data from extreme environments.

Thank you for acknowledging the contribution of our work to DNRA and understanding the promiscuity of nitrite reduction. We appreciate the excitement about the new directions of research that this type of study can support.

There are a few places in this manuscript that could benefit from some minor revisions.

We will go point-by-point to discuss the points raised by the reviewer. Thanks for the detailed feedback to help us improve the manuscript.

- Line 89: missing word, nitrite as a free ...

Original: "nitrite as free intermediate"

Revised: "nitrite as a free intermediate"

- Line 90: missing word, as an intermediate...

Original: "nitrite as free intermediate"

Revised: "nitrite as a free intermediate"

- Line 101: missing NasB. In line 87 the authors specifically mention NasB as a possible assimilatory nitrite reductase, but don't add it to Table 1. The genome of *A. acetoxydans* does not encode an annotated NasB. The assimilatory nitrite reductases should be mentioned not just the dissimilatory ones.

Original line 101: it does not encode any of the six known nitrite reductases (NrfAH, NirA, NirB, HaoA, ONR and OTR)

Revised line 101: it does not encode any of the seven known nitrite reductases (NrfAH, NirA, NirB, nasB HaoA, ONR and OTR).

Selected nitrite reductases in table 1 are based on the data regarding the hypothesized pathway as observed in this study. Considering the promiscuity of sulfite/nitrite reductases, the other nitrite reductases, of which we have no in-depth data in (a)SRB would not contribute to the homology search. Therefore not all nitrite reductases mentioned in the introduction are part of this table. We opted to include NrfA and NapA to give an indication of nitrate reduction as a NarG was detected in this study.

- Figure 1: I had trouble distinguishing the colors here. The nitrite and nitric oxide lines are indistinguishable.

Thanks for sharing, we adjusted the colour of nitric oxide. Now this should be better distinguishable

- Line 146: Supplemental figure S2, graph B. It seems to be missing the nitrite line in the graph.

In this experiment (verification of the reduction of nitrite to ammonium) no nitrite is measured, as experimental constraints prohibited to measure, next to ammonium and hydroxylamine, nitrite simultaneously.

Chemical controls in this and other experiments did not show degradation of nitrite to the order of magnitude (mM) shown here we (overnight in the μ M range whereas this experiment was in a 4.5 hour timespan).

- Line 160: Its super interesting that the sulfate reduction pathways are constitutively expressed. This is another valuable contribution of this manuscript to understanding the physiology of these acidophilic microbes.

We appreciate and share the enthusiasm.

- Figure 2: I don't understand the figure color explanation. Yellow is 2 and purple is -2. How does this relate to the legend text (line 848) of normalized counts and LFQ values?

Thanks for sharing, we adjusted the figure legend to clarify.

Original: “**Figure 2.** Heatmaps showing the *A. acetoxydans* comparative transcriptomics with normalized counts (left) and proteomics with LFQ values (right) comparing nitrate (N1-N4) to sulfate (S1-S4) reduction conditions. (...)”

Revised: “**Figure 2.** Heatmaps showing the *A. acetoxydans* comparative transcriptomics with normalized counts (left) and proteomics with LFQ values (right) comparing nitrate (N1-N4) to sulfate (S1-S4) reduction conditions. Heatmap color codes represent z-score normalized values per row comparing across reduction conditions. (...)”

- Line 186: I like this paragraph of description for DEACI_1836. It fits well with the discussion (Line 328)

Thank you.

- Line 205: word choice, I am not clear what did not change expression levels.

Original: “However, it did not change in expression level, nor was it detected in the proteome.”

Revised: “However, *norBC* did not change expression level and neither was *NorBC* detected in the proteome.”

- Line 261: According the Table 1, the authors show that *T. narugense* does have a NirB homolog. Could this assimilatory nitrite reductase be used for ammonification?

Short answer: yes, this is a highly likely candidate and we adjusted the sentence accordingly to put more caution into it. The promiscuity in nitrite/sulfite reductases makes contributing functions without further study difficult.

We adjusted the sentence accordingly.

Original: “Only *T. narugense* in this genome comparison is reported as being able to perform nitrite reduction without encoding AsrABC, or any other known nitrite reductase.”

Revised: “Only *T. narugense* in this genome comparison is reported as being able to perform nitrite reduction without encoding AsrABC, but does encode a putative *nirB*.”

- Line 265: According to Table 1, *D. hafniense* does have a Hcp, but does not have DEACI_1836 homolog.

Based on the adjustments in the next segments this sentence was removed.

- Table 1:

Line 863: Add *Thermodesulfobium* as species mentioned in the legend text

For Table 1 the genera are named for the closest relatives to the genus *Acididesulfobacillus*, although the genus *Thermodesulfobium* has aSRB, the genus does not have neutrophiles and despite containing aSRB the genus is not closely related to *Acididesulfobacillus*. Therefore, the genus is not named in the following sentence: “Prevalence of identified candidates for DNRA in characterized aSRB and neutrophilic type strains of the closest genera to *Acididesulfobacillus*: *Desulfosporosinus* and *Desulfitobacterium*.” As these are phylogenetically the closest related.

For clarity the description of Table 1 is adjusted

Original: “Table 1. Prevalence of identified candidates for DNRA in characterized aSRB and neutrophilic type strains of the closest genera to *Acididesulfobacillus*: *Desulfosporosinus* and *Desulfitobacterium*. The neutrophilic type strains are depicted in bold. Nitrate and nitrite reduction indicate reported ability to reduce the compound. Identity scores higher than 30% are listed. For *Ds. meridiei*, the initial characterisation paper did not observe nitrate reduction whereas later on nitrate reduction is reported (22). Reference sequences were taken from ¹*Escherichia coli* strain K12 and ²*Paracoccus denitrificans*.”

Revised: “Table 1. Prevalence of identified candidates for DNRA in characterized aSRB. Of the closest genera to *Acididesulfobacillus*: *Desulfosporosinus* and *Desulfitobacterium*, the neutrophilic type strains are included (depicted in bold). ... “

I noticed one mistake, *D. hafniense* strain DCB-2 T does encode a NosZ protein (also line 267). Perhaps it is less than 30% similar to *Paracoccus*, which is why it wasn't included here. Its worth checking.

Thanks for noticing. We rechecked carefully, and unfortunately something went wrong with importing through the command line from NCBI for genes that have the MULTISPECIES annotation and only have the [genus] identifier rather than the [genus species]. We redid the table manually (manual selection of the genome and protein files, followed by manual BlastP) to ensure everything is now correct. For increased accuracy we also adjusted the total score to > 100. Based on this some changes were made to the table. Among these changes is indeed a NosZ (and, therefore, DEACI_0711) for *D. hafniense*.

In light with the minor adjustments in the table, this led to the following text adjustments in the results section of Table 1.

Original: “An identity score ≥ 30 was implemented as cut-off criterium”

Revised: “An identity score $\geq 30\%$ and total score ≥ 100 were implemented as cut-off criteria”

Original: "DEACI_1836, was identified with a >70% identity score in most of the strains"

Revised: "DEACI_1836, was identified in all strains, with a > 70% identity score in most of the strains."

Due to this revision and minor table adjustment the following line was removed:

Original: "The DEACI_1836 of *A. acetoxydans* was detected in all strains except *Db. hafniense*"

Original: "In contrast to the high Hcp prevalence, NosZ was only detected in *Ds. meridiei*, *Ds. youngiae* and *Db. chlororespirans*."

Revised: "In contrast to the high Hcp prevalence, NosZ was only detected in *Ds. meridiei*, *Ds. youngiae*, *Ds. dehalogenans*, *Ds. hafniense* and *Db. chlororespirans*."

- Line 300: I think this is well-reasoned. Nitrous oxide production, if any, was probably very low and was likely converted to nitrogen gas quickly.

Thank you for the kind words.

- Line 319: Again, I like how the authors leave this as a very cautious hypothesis.

Thank you for the kind words.

- Line 329: missing word, to a described NirA

Original: "to described NirA"

Revised: "to described NirA sequences"

- Line 332: I like that the authors show this alignment. It is rough, but key active site domains are conserved.

Thanks for your remark.

- Line 338: word choice, please clarify what you mean by "both conditions"

Original: "Pfor is highly abundant in both conditions"

Revised: "Pfor is highly abundant in both sulfate and nitrate reducing conditions"

- Line 356: missing word, As a nitric oxide...

Original: "As nitric oxide reductase..."

Revised: "As a nitric oxide reductase..."

- Line 358: word choice, "factor"

Original: "Hcp is a major factor for handling"

Revised: "Hcp is a major contributing protein for handling"

- Line 396: word choice, encoding a known nitrite reductase...

Original: "Although SRB encoding a nitrite reductase are more numerous"

Revised: "Although SRB encoding a known nitrite reductase are more numerous"

- Line 406: what do the authors mean by "widespread"?

We adjusted the text to make the message more clear.

Original: "currently described NirA are incredibly widespread complicating functional interpretation"

Revised: "currently described NirA are from phylogenetically very distinct organisms, which complicates functional interpretation"

- Line 409: According to Table 1, AsrC is detected in neutrophiles, not AsrA.

We adjusted the text to make the message more clear as this statement does not refer to the table.

Original: "However, AsrA is also detected in neutrophiles hence this trait cannot be ascribed to being acidophilic or isolated from acid mine drainage environments"

Revised: "However, AsrA is also detected in neutrophiles, such as *E. coli* and *S. typhimurium*, hence this trait cannot be ascribed to being acidophilic or isolated from acid mine drainage environments"

- Line 412: Hcp is detected in all examined (a)SRB and neutrophilic SRB as well. I'm kind of getting mixed up in this paragraph about the comparisons between (a)SRB and all SRB. Given the next sentence about using Hcp as a marker, perhaps the authors want to stay focused just on the (a) SRB.

We streamlined the text according to your suggestion.

Original: "Aside from nitrite reduction, the most crucial step in tackling nitrosative stress is performed by Hcp which was detected in all checked aSRB and is widely distributed among both facultative and strictly anaerobic microorganisms"

Revised: "Aside from nitrite reduction, the most crucial step in tackling nitrosative stress is performed by Hcp which was detected in all checked (a)SRB"

- Line 425: missing word, The medium is a modification...

Thank you, adjusted

Original: "Medium is a modified version of the medium used by..."

Revised: "The medium is a modification of the medium used by..."

- Line 481: missing comma, In technical duplicates, the absorbance...

Thank you, adjusted.

- Line 576: I could not access the RNA-seq data using this accession number or the organism name.

We apologize for caused inconvenience, although uploaded to the ENA database their status was still listed as "Private". We changed the status from "Private" to "Open". After doing so we rechecked whether they were now available and that was the case. Thus all reviewers (and others) should now be able to access the RNAseq data.

Details:

Link: <https://www.ebi.ac.uk/ena/>

Accession: PRJEB64595

Title: DNRA Acididesulfobacillus acetoxydans

- Line 578: I could not access the proteomics data using this this dataset identifier or the organism name

We apologize for caused inconvenience, once accepted for publication we will directly make the dataset publicly available.

For the reviewers the data is available:

Link: <https://www.ebi.ac.uk/pride/>

Username: reviewer_pxd045135@ebi.ac.uk

Password: FdcngZSj

Project accession: PXD045135

Reviewer #2

The authors report their recent discovery of a DNRA phenotype in a rather unexpected acidophilic bacterium without *nrfA* or *nirB*, the previously known nitrite reductase genes, in its genome. The manuscript is very well-written in general, and the experimental designs and procedures and the results all appear sound. I would really appreciate the authors' adherence to strong scientific fundamentals in preparing their manuscript, which made the review process a very pleasant process.

Thank you for the constructive and diligent review, we appreciate your kind words and appreciation of the manuscript.

I have only two major concerns: 1) the authors' negligence to the recent publications in ASM journals that would provide very strong supports to the findings reported in this current study (for example, Heo et al., 2020 AEM paper and Yoon et al., 2023 mBio paper), and 2) heavy reliance on the negative proteomics data for hypothesizing the enzymes pertaining to DNRA metabolism. Proteomics, although immensely popular today, has many drawbacks: that the results can never be discussed quantitatively, and that membrane proteins cannot be reliably detected. I have several additional minor comments that may help the authors improve the quality of their manuscript.

We appreciate the shown concern and in the following section provide a point-by-point answer to your suggestions. We appreciate and recognize your concerns and, with this in mind addressed various sections accordingly. We like to express our gratitude for the provided literature which improved the manuscript and acknowledge the provided service by the reviewer in this peer-review process.

- Line 188 - 193: I would only discuss about the two domains which have predicted functions. A 'domain of unknown function' carries no important information.

We adjusted according to your suggestion:

Original: "DEACI_1836 encodes a 300 amino acid containing protein with 3 domains: a domain of unknown function (PF08984), a nitrite- and sulfite-reductase ferredoxin-like half domain (PF03460) and a nitrite- and sulfite-reductase [4Fe4S] domain (PF01077)."

Revised: "DEACI_1836 encodes a 300 amino acid containing protein with two known domains: a nitrite- and sulfite-reductase ferredoxin-like half domain (PF03460) and a nitrite- and sulfite-reductase [4Fe4S] domain (PF01077)."

- Line 199 - 200: The Yoon et al., 2023 mBio paper may help explain the presence of *nosZ* in the genome and the purpose of its expression to the microorganism.

We thank the reviewer for sharing this paper and incorporated the Yoon *et al.*, 2023 paper in the discussion when *NosZ* is discussed.

Original: "(...)This aligns with the unexpected upregulation of the operon encoding the nitrous oxide reductase *NosZ* and ABC transporter *NosDFY* required for *NosZ* formation (63, 64). Nitrous oxide is subsequently reduced to dinitrogen gas by *NosZ* with electrons (...)"

Revised: "(...)This aligns with the unexpected upregulation of the operon encoding the nitrous oxide reductase *NosZ* and ABC transporter *NosDFY* required for *NosZ* formation (63, 64). Increased nitrous oxide concentrations were shown to hamper DNRA upon oxic-to-anoxic transition in *Bacillus* sp. DNRA, which could be alleviated by timely activated *NosZ* (Yoon *et al.*, 2023). In a similar fashion the build-up of nitrous oxide from nitric oxide

can also hamper DNRA activity in *A. acetoxydans*, explaining the role of NosZ during DNRA (Yoon *et al.*, 2023). Hence, produced nitrous oxide is subsequently reduced to dinitrogen gas by NosZ with electrons (...)"

- Line 227 -228: This is odd. Possibly DEACI_3264 may be a structural protein.

That could indeed be the case.

- Line 208 -232: This entire section is a discussion of why some of the genes possibly involved in N cycling are not likely as candidates for DNRA or nitric oxide reduction. Most of these are reiterated in the discussion section. I would recommend summarizing this section into a supplementary table and remove it.

Below the streamlined section with a reference to a supplementary file, where the more descriptive text is placed including a table per reviewer' suggestion.

Revised text: "Two gene clusters, DEACI_2560-2564 and DEACI_3268-3271, were upregulated approximately thirteen-fold when nitrate was used as final electron acceptor (Supplementary File S4). The annotation of either set of genes was not clear, but both operons encode NrfD-like subunit-containing molybdopterin oxidoreductase complexes, which are proteins involved in ion translocation. Functional annotation of these oxidoreductases is difficult considering the metabolic versatility of the oxidoreductases and modularity of their subunits (18, 19). They are not identified as nitrite reductase candidates given their lack of distinct homology to any nitrite reductase, nor are their expression and abundance levels equal to the identified candidates."

- Line 247 - 268: This section belongs to the discussion section. As this manuscript is a bit too long for what it contains, I would recommend combining the result and discussion sections. That would remove redundancies and help the readers' focus on the key findings.

We streamlined this section by removing the discussion-like parts.

Original: "Transporters for identified and potential intermediates were detected: uptake of nitrate and export of formed nitrite was likely performed by the nitrate-nitrite transporter NarK (DEACI_1355). Interestingly, a sulfate permease protein (DEACI_0281) was seventeen-fold upregulated under nitrate reducing conditions to 605 TPM (although it was not identified in the proteome). For the three potential intermediates, nitric oxide, nitrous oxide, and hydroxylamine, no dedicated transporters were encoded. Formed ammonia, and potentially hydroxylamine, might be exported through the ammonia transporter Amt (DEACI_0023). Upstream there were two highly upregulated hypothetical protein encoding genes, DEACI_0021 and DEACI_0022, which increased thirteen- and sixteen-fold respectively. Only DEACI_0023 was detected in the proteome, where it significantly decreased under nitrate reducing conditions."

Revised: "Uptake of nitrate and export of formed nitrite was likely performed by the nitrate-nitrite transporter NarK (DEACI_1355). For the three potential intermediates, nitric oxide, nitrous oxide, and hydroxylamine, no dedicated transporters were encoded. Formed ammonia, and potentially hydroxylamine, might be exported through the ammonia transporter Amt (DEACI_0023). DEACI_0023 was detected in the proteome, where it significantly decreased under nitrate reducing conditions."

On the comment of combining results and discussion, we do not think that it benefits the manuscript and additionally it is not allowed in this journal.

- Line 289: The sentence that starts with "In this chemical controls..." is a bold statement, unless NO formation is stoichiometric to NO₂⁻ loss. NO is a very difficult substance to hold stable, even just with pure water and a very small amount of oxygen penetrating into the system.

We made our statement more cautious as no decrease in NO was observed.

Original: "In these chemical controls, no loss of nitric oxide.."

Revised: "In these chemical controls, no decrease of nitric oxide.."

- Line 313: Whenever the authors bring up the previous studies on the orthologues of the genes / enzymes in other organisms, they should mention about the similarity of the enzymes in those organisms from that in *Acididesulfobacillus acetoxydans*.

We adjusted the text accordingly by providing identity scores (%).

Original: "Interestingly, the *asrABC* deletion mutants of *Salmonella typhimurium* were still capable of nitrite reduction"

Revised: "Interestingly, the *asrABC* deletion mutants of *Salmonella typhimurium* (43%, 40% and 42% protein identity scores to *A. acetoxydans*, respectively) were still capable of nitrite reduction"

Original: "the increased sensitivity to nitrosative stress of Hcp mutants of *S. typhimurium*, *Porphyromonas gingivalis* and *Desulfovibrio gigas*"

Revised: "the increased sensitivity to nitrosative stress of Hcp mutants of *S. typhimurium*, *Porphyromonas gingivalis* and *Desulfovibrio gigas* (42%, 65% and 48% protein identity scores to *A. acetoxydans*, respectively)"

Original: "*Desulfovibrio desulfuricans* which showed increased expression of *hcp* (...)"

Revised: "*Desulfovibrio desulfuricans* which showed increased expression of *hcp* (69% protein identity score to *A. acetoxydans*) (...)"

For another NosZ instance see previous comment above. For other instances the similarity can be found in Table 1. For our NirA homology to DEACI_1836, we believe the supplementary file S4 gives adequate information.

- Line 365: This is one of the examples, where referring to the Heo et al., 2021 paper would be very helpful.

We incorporated this paper

Original: "NorBC was hardly expressed in the transcriptome and did not change expression level."

Revised: "NorBC was hardly expressed in the transcriptome and did not change expression level, potentially the expression of NorB was affected by active DNRA (#)"

- Line 371: This statement is an over-simplification. NosZ can receive electrons from many different electron donor compounds via varying electron transfer pathways.

We adjusted this line to ensure it is not over-simplified.

Original: "(...) by NosZ with electrons supplemented from PetABC"

Revised: “(...) by NosZ with electrons supplemented from PetABC, although additional electron transfer pathways might be involved”

- Line 381: Einsle et al., 2011 Methods in Enzymology paper indicates that NO may be intermediates of NrFA, and Heo et al., 2021 paper and Yoon et al., 2023 paper support the possibility with physiological data. Although NO does not get detected extracellularly with the analytical equipment used by the authors, it is highly likely that NO is one of the intermediates of this enigmatic DNRA pathway.

The role of NO in DNRA remains indeed enigmatic as the reviewer stated. However this study for this particular DNRA pathway does not provide evidence for the role of NO as intermediate. Perhaps it is, but we cannot claim this from our data and saying so, in our opinion, is too speculative.

- Line 388: This hypothesis is too much of a stretch. It is highly unlikely that nitrite can be converted to NO via NarG activity.

We appreciate and share the caution of the reviewer, however based on this study we cannot prove or disprove nor is stating likeliness based on our data relevant. Therefore every time this hypothesis is suggested we ensured to do so cautiously. This is also displayed in Figure 3 and the legend.

However, it is not too much of a stretch, as the production of NO from nitrite through NarG has been proven, and adequate references are provided in the discussion.

- Line 401: By definition, 'Denitrification' is reduction of dissolved nitrogen species (NO₃⁻ and/or NO₂⁻) to gaseous species (NO, N₂O, and N₂). I would be very careful to term NO-to-N₂O reduction as 'partial denitrification'.

We appreciate the caution of the reviewer and adjusted the two instances of “partial denitrification”.

Original A: “Hence, this partial denitrification flux seems to be minor in terms of nitrogen flux, yet important to maintain viability of *A. acetoxydans* during nitrate reduction.”

Revised A: “Hence, this reductive nitric oxide flux seems to be minor in terms of nitrogen flux, yet important to maintain viability of *A. acetoxydans* during nitrate reduction.”

Original B: “Based on the performed genome comparison, the combination of the projected DNRA and denitrification pathway is only present in *A. acetoxydans* with the partial denitrification route from nitric oxide to dinitrogen gas only conserved in a few SRB.”

Revised B: “Based on the performed genome comparison, the combination of the projected DNRA and denitrification pathway is only present in *A. acetoxydans*. The reductive route from nitric oxide to dinitrogen gas is only conserved in a few SRB, none of which acidophilic.”

- Line 402-404: This sentence needs to be rewritten with better clarity.

Thanks for sharing, we incorporated this in the comment above.

- Line 409-410: If AsrA is a periplasmic or cytoplasmic enzyme, the environmental pH should not be an important factor in its possession, expression, or activity.

We disagree, to illustrate, many pH homeostasis mechanisms resolve around cytoplasmic enzymes and the environmental pH impacts their expression and activity.

In the same manner, the importance/role of AsrA can differ based on environmental pH despite being a cytoplasmic enzyme. It is good to note that we do not make a bold claim here as in the following sentences we already illustrate the prevalence of AsrABC in neutrophiles and say it is not an acidophilic trait. However, as the remainder of the text indicates we noted that if AsrA is instead of a sulfite reductase functioning as a nitrite reductase this could in aSRB play a different role than in neutrophiles.

- Line 413-415: I strongly disagree with this statement. Hcp is too wide-spread to be used as a marker gene for DNRA.

Correct, we agree. Hcp being a marker for DNRA was not the intended message of this sentence. Rather the potential role/importance of Hcp for handling nitrosative stress making it an interesting target for further study in low pH environments. We adjusted this line.

Original: "Given the chemical disproportionation of nitrite into nitric oxide in anoxic acidic environments, perhaps Hcp is advantageous in handling nitrosative stress making it an interesting marker gene for further study in anoxic acidic conditions"

Revised: "Given the chemical disproportionation of nitrite into nitric oxide in anoxic acidic environments, perhaps Hcp is advantageous for handling nitrosative stress. The role of Hcp in anoxic acidic conditions is an interesting target for microbial physiology studies."

- Line 431: Why were alkaline trace elements used for an acidic medium?

This is a technical term, to create a stock solution of these trace elements an alkaline condition is required for solubilization. Hence, the stock solution used to prepare the medium is termed alkaline trace element solution. The nomenclature is kept to be consistent with the reference for the medium provided.

- Line 436: 0.1 g/L yeast extract may be a serious problem here. The nitrogen content of a typical yeast extract powder is around 10%, and a substantial portion of this nitrogen may be mineralized to NH₄⁺. Perhaps, this was why the authors supplemented their results with the resting cell experiments. The authors should mention this in the methods or result section.

Thanks for addressing this.

To respond to the raised issue: we believe this does not impact the story as in each experiment with yeast extract the nitrogen intermediates were measured with nitrogen balances closing. The resting cell experiments were indeed not supplemented with yeast extract as these were particularly aimed at identifying DNRA intermediates in which yeast extract could indeed hinder observations. However all obtained experimental results aligned, with or without yeast extract.

We recognize the need for mentioning it and adjusted the following section in the results section. On purpose we incorporated this in the first paragraph before description of chemical controls. This to ensure the reader gets the information straight away.

Original: "No nitrogen loss was observed at any point in the cultivation, excluding additional end products. (...)"

Revised: "No nitrogen loss was observed at any point in the cultivation, excluding additional end products. In these cultivations yeast extract (0.1 g L⁻¹) was present, which contains nitrogenous compounds yet no interference was detected on the nitrogen balances. (...)"

- Line 488: I doubt that concentration of dissolved NO was measured. The detection limit should be presented in terms of the gaseous concentration.

You are correct, we adjusted the detection limit accordingly to parts per million (ppm).

- Line 502-505: It appears that some important details are missing for the RNASeq procedure. It is often unnecessary to list all the details for extraction, processing, and sequencing, but considering all the minor details presented, some more detail would be needed here.

Thanks for acknowledging the detailed materials and methods. We find it important that enough details are provided to redo the experiment based on the materials and methods section.

For the particular section the reviewer commented on, we believe these details suffice to rerun this experiment as the rRNA Depletion Kit (Illumina Ribo-Zero ...), instrument platform (Illumina), sequencer model (NovaSeq6000), sequencing company (Novogene), library strategy (RNAseq), and library layout (paired) are provided.

- Line 516: This unit TPM is not normalized by the gene length. Large genes will have higher TPM values.

Thank you for addressing this point. There is sometimes confusion on the different nomenclature on TPM, RPKM, FPKM, and the normalization of RNAseq data overall. To clarify, inherently TPM does normalize for gene length, but not *just* for gene length as it also normalizes for sequencing depth.

Way of TPM calculation:

1. Divide the read counts by the length of each gene in kilobases, this gives the reads per kilobase (RPK). Therefore the data is normalized by gene length, but not *just* by gene length.
2. Sum all RPK values per sample
3. Divide this by 1000000 (one million); this is the per million scaling factor from which TPM derives its name and allows to normalize for gene length and sequencing depth.

Overall, this is irrelevant for the differential expression analysis, TPM is a way to give a normalized of obtained counts. We hope this clarifies the selection and our rationale for using TPM. In line with your previous comment, we elaborated here as well to maintain consistency.

Thanks again for helping us improve our manuscript, the attention dedicated to your rigorous review is something we highly appreciate.

Reviewer #3

In this study, authors analyzed the pathway of dissimilatory nitrate reduction to ammonium (DNRA) in *Acidithiobacillus acetoxidans* physiology test. With comparative analysis of transcriptomics and proteomics, authors found that the NADH-linked sulfite reductase AsrABC and/or a putatively ferredoxin-dependent homolog of the nitrite reductase NirA (DEACI_1836) are the key enzyme(s) of nitrite reduction to ammonia. This work extended the knowledge of DNRA especially at extreme environments.

The topic is interesting, the conclusion is supported with solid results. The manuscript is well written.

Thank you for recognizing the contribution of our work to increasing the understanding of extreme environments such as AMD. We appreciate your kind words and attention to detail, thanks for helping us improve the manuscript.

Minor comments:

- In Fig.1, I think what you determined is ammonium not ammonia. Please carefully differentiate the both.

Thanks for your comment. We adjusted the Y-axis, legend, description and carefully went over the text to correct where needed. This is also adjusted in all other Figures and Figure legends, including the supplementary.

- In Supplementary Table 1, "NO-" should be changed to "NO₂-"; "glycerol" and "acetate" should be changed to "glycerol [mM]" and "acetate [mM]", respectively. Furthermore, the data of glycerol is not matched with Fig. S1, please check it.

Thanks for your diligent attention to detail on the Supplementary material. We corrected the following:

Original: "NO⁻ [mM]", "Glycerol", "Acetate"

Revised: "NO₂⁻ [mM]", "Glycerol [mM]", "Acetate [mM]"

Thanks for addressing the glycerol concentration. After carefully rechecking the glycerol data we confirm that the glycerol data of Supplementary Table 1 fits the glycerol data depicted in Fig S1. Perhaps some confusion arose as the secondary Y-axis refers to the metabolite concentration [mM].

Re: mSystems00967-23R1 (A novel mechanism for dissimilatory nitrate reduction to ammonium in *Acidithiobacillus acetoxidans*)

Dear Dr. Irene Sánchez-Andrea:

Your manuscript has been accepted, and I am forwarding it to the ASM production staff for publication. Your paper will first be checked to make sure all elements meet the technical requirements. ASM staff will contact you if anything needs to be revised before copyediting and production can begin. Otherwise, you will be notified when your proofs are ready to be viewed.

Featured Image Submissions: If you would like to submit a potential Featured Image, please email a file and a short legend to mSystems@asmusa.org. Please note that we can only consider images that (i) the authors created or own and (ii) have not been previously published. By submitting, you agree that the image can be used under the same terms as the published article. File requirements: square dimensions (4" x 4"), 300 dpi resolution, RGB colorspace, TIF file format.

Sincerely,
Liyuan Ma
Editor
mSystems

Reviewer #1 (Comments for the Author):

For the manuscript, "A novel mechanisms for dissimilatory nitrate reduction to ammonium in *Acidithiobacillus acetoxidans*", the authors do a good job of addressing my minor revision recommendations. I was able to access the EBI files of the RNAseq and proteome files. I appreciate their response to my comment about *nosZ* in *D. hafniense* and their manual search to confirm presence or absence of these genes in Table 1. It's a really good manuscript and a good contribution.

Reviewer #2 (Comments for the Author):

The authors have addressed the comments well. The only few things that may be better to change is:

Line 227-228: I doubt there are dedicated transporters for NO, N₂O, and NH₂OH, as these non-ionic compounds may diffuse relatively freely through the membrane.

12-23-23

Response to reviewed manuscript

For the manuscript, "A novel mechanisms for dissimilatory nitrate reduction to ammonium in *Acidithiobacillus acetoxidans*", the authors do a good job of addressing my minor revision recommendations. I was able to access the EBI files of the RNAseq and proteome files. I appreciate their response to my comment about *nosZ* in *D. hafniense* and their manual search to confirm presence or absence of these genes in Table 1. It's a really good manuscript and a good contribution.